# Dementia with Lewy bodies post-mortem brains reveal differentially methylated CpG sites with biomarker potential

Xiaojian Shao [1✉], Sangeetha Vishweswaraiah [2], Miroslava Čuperlović-Culf[1,3,4], Ali Yilmaz[2,5],
Celia M. T. Greenwood [6,7,8], Anuradha Surendra[1], Bernadette McGuinness[9], Peter Passmore[9],
Patrick G. Kehoe [10], Michael E. Maddens[2,5], Steffany A. L. Bennett[3,4], Brian D. Green[11],
Uppala Radhakrishna [2,5] & Stewart F. Graham [2,5✉]

Dementia with Lewy bodies (DLB) is a common form of dementia with known genetic and environmental interactions. However, the underlying epigenetic mechanisms which reflect these gene-environment interactions are poorly studied. Herein, we measure genome-wide DNA methylation profiles of post-mortem brain tissue (Broadmann area 7) from 15 pathologically confirmed DLB brains and compare them with 16 cognitively normal controls using Illumina MethylationEPIC arrays. We identify 17 significantly differentially methylated CpGs (DMCs) and 17 differentially methylated regions (DMRs) between the groups. The DMCs are mainly located at the CpG islands, promoter and first exon regions. Genes associated with the DMCs are linked to "Parkinson's disease" and "metabolic pathway", as well as the diseases of "severe intellectual disability" and "mood disorders". Overall, our study highlights previously unreported DMCs offering insights into DLB pathogenesis with the possibility that some of these could be used as biomarkers of DLB in the future.

[1] National Research Council of Canada, Digital Technologies Research Centre, Ottawa, Canada. [2] Oakland University-William Beaumont School of Medicine, Rochester, MI 48309, USA. [3] Ottawa Institute of Systems Biology, Ottawa, Ontario, Canada. [4] Department of Biochemistry, Microbiology, sand Immunology, Faculty of Medicine, University of Ottawa, Ottawa, Ontario, Canada. [5] Beaumont Research Institute, Royal Oak, MI 48073, USA. [6] Lady Davis Institute for Medical Research, Jewish General Hospital, Montréal, Canada. [7] Department of Epidemiology, Biostatistics and Occupational Health, McGill University, Montréal, Canada. [8] Gerald Bronfman Department of Oncology, McGill University, Montréal, Canada. [9] Centre for Public Health, School of Medicine, Dentistry and Biomedical Sciences, Queen's University Belfast, Belfast, UK. [10] Dementia Research Group, Translational Health Sciences, Bristol Medical School, University of Bristol, Bristol, UK. [11] Institute for Global Food Security, School of Biological Sciences, Faculty of Medicine, Health and Life Sciences, Queen's University Belfast, Northern Ireland, UK. ✉email: xiaojian.shao@nrc-cnrc.gc.ca; stewart.graham@beaumont.edu

Dementia with Lewy bodies (DLB) is an age-associated neurodegenerative disease that can be clinically characterized by cognitive fluctuations, extrapyramidal motor symptoms and early visual hallucinations[1,2]. DLB accounts for 10-15% of all pathologically defined dementias and is the second most common type of degenerative/progressive dementia after Alzheimer's disease (AD)[3–5]. Neuropathologically, DLB is characterized by the abnormal accumulation and deposition of misfolded and aggregated α-synuclein (α-syn) in the brain and nervous system, causing neuronal cell death and synaptic dysfunction[5–7]. α-synuclein found in DLB-affected brain tissues is initially identified in cortical areas. This protein propagates through the olfactory system disrupting the limbic regions and manifesting in the neocortical areas of the brain[8]. It shares clinical, genetic, and pathological features with Parkinson's disease (PD) and AD, which can make a precise diagnosis more difficult. As with most other types of dementia, there are no available treatments, which stop or slow the progression of DLB. Compared with other dementias, DLB is also believed to have an earlier age of onset, and a shorter disease course with a lower survival rate[9,10]. Therefore, there is an urgent need to understand the etiopathophysiology of the disease to help develop effective disease-modifying therapies.

Currently, the exact cause of DLB is still largely unknown, but it has been suggested that both genetic and environmental factors contribute to the disease pathogenesis[8,11–17]. Particularly, studies have addressed DNA methylation of *SNCA* and α-syn conformational change on exposure to environmental toxins such as pesticides and heavy metals (ex: manganese)[16,17]. In an in-vitro cellular model study, toxin-like 1-methyl-4-phenylpyridinium was observed to demethylate the *SNCA* promoter leading to overexpression of *SNCA* that may contribute to the formation of Lewy bodies[8,18]. Epigenetic mechanisms including the methylation of CpG dinucleotides on genomic DNA are known to mediate the gene-environment interactions, and are thought to be involved in the development of the disease. Such mechanisms regulate the transcriptomes of neuronal cells, and play important roles in neurodegeneration[19–21]. For instance, Humphries et al. integrated whole transcriptome and DNA methylation analysis and identified disruptions in both DNA methylation and transcription with genes in the myelination network which related to synaptic function and behavioral response in both late-onset Alzheimer's disease (LOAD) and DLB[22]. Compared with other neurodegenerative diseases such as PD and AD, where comprehensive studies of DNA methylation dynamics have been conducted, very few studies have been undertaken for DLB[23,24]. As α-syn aggregation is the predominant pathology associated with the disease, the methylation of the *SNCA* has been the primary area of research interest. Desplats et al. found global DNA hypomethylation and highlighted the hypo-methylation of intron 1 of the *SNCA* gene in post-mortem brain tissue from patients who died from DLB[25]. These findings were later validated by Funahashi et al. who studied 10 CpGs on intron 1 of the *SNCA* gene in peripheral blood[26]. Tsuchida et al. reported that the gene body of *FGFR3*, the upstream gene of the mitogen-activated protein kinase (MAPK) pathway, which is normally suppressed when α-syn is overexpressed, was hyper-methylated in Lewy body disease (LBD) brains including the DLB PD, and PD with dementia (PDD)[27]. Recently, Ozaki et al. detailed increased leukocyte CpG methylations in *DRD2* in DLB patients[28]. Moreover, Sanchez-Mut et al. identified 1075 differentially methylated promoters in LBD related to changes in gene expression[29]. Interestingly, they observed differential methylation of *ANK1*, a gene previously reported to be associated with AD[29]. These studies represent the early efforts to explore DNA methylation sites in DLB. These studies are extremely limited because they are either based on

targeted candidate genes or were only analyzed on couple of samples. However, they do highlight the necessity for larger studies investigating genome-wide DNA methylation alternations associated with DLB.

Very recently and during the preparation of our study, Pihlstrøm et al.[30] performed a large epigenome-wide association study (EWAS) in a cohort of DLB post-mortem human brain tissue (frontal-cortex; $n = 332$) to identify DNA methylation changes associated with the disease. Like Pihlstrøm et al., this study aims to demonstrate the power of profiling genome-wide DNA methylation sites in pathologically confirmed cases of DLB but focusing on a different brain region. Specifically, profiling was undertaken in post-mortem brain tissue (parietal cortex, Broadmann area 7) from 15 DLB cases and compared with 16 cognitively normal control subjects using Illumina MethylationEPIC array. The identified sites and regions were revealed to be associated with Parkinson's disease pathway, diseases of severe intellectual disability and mood disorders using functional enrichment analysis. Weighted gene co-expression network analysis (WGCNA)[31,32] showed that DNA methylation at different genomic locations (of different genes) could have been co-regulated in association with DLB. Some of these co-regulated modules, for example, the greenyellow module, showed enrichment in the biological process of positive regulation of GTPase activity and regulation of RhoA activity. It also demonstrated significant association with diseases of Hypoventilation, Lewy Body Disease and neuromuscular diseases. Overall, our analyses provide insights into the epigenetic architecture of DLB and catalogues for the underlying DNA methylation changes that occur in the etiopathophysiology of DLB. There is a potential for the development of biomarker panels for the DLB diagnosis which requires clinical studies.

## Results

The results of student's t-tests showed that the DLB and control groups are similar on age (79.6 ± 6.5 versus 81.9 ± 7.95; $p$-value = 0.39), post-mortem interval (PMI) (36.71 ± 19.86 versus 42.08 ± 18.44; $p$-value = 0.44) and neuronal proportions (0.73 ± 0.07 vs. 0.72 ± 0.03; $p$-value = 0.91 for NeuN_neg, and 0.29 ± 0.06 vs. 0.31 ± 0.04; $p$-value = 0.39 for NeuN_pos). When breaking down the two groups by sex, one-way ANOVA analysis showed that age, PMI and neuronal proportions were not statistically different across the four groups (Table 1). This indicates that pathology (i.e., Braak stage) scoring of each participant can be confidently associated with the measured DNA methylation profiles.

We first performed a series quality control filters on the IIlumina's EPIC probes (see the Method for the details). We started with 865,859 CpG probes that are mappable to the human genome (version hg19) based on the EPIC annotation file (R package: IlluminaHumanMethylationEPICanno.ilm10b4.hg19, version 10.b4), 10,841 poor quality probes (i.e., detection $p$-value is not less than 0.01 in all samples) were removed. 28,260 probes were further removed as they were affected by SNPs (either at the CpG interrogation site or at the single nucleotide extension). In addition, 40,565 and 2,576 probes were also excluded that demonstrated cross-reactivities and overlapped with non-CpG context (i.e. CpA, CpC and CpT). Finally, 17,209 probes located on the sex chromosomes were deleted. Subsequently, 766,399 CpGs were used for downstream analysis after removing an additional 9 CpGs which contained missing data. Of note, no samples were removed due to quality concerns (Supplementary Fig. 1a). Furthermore, the sex chromosome probes were used to estimate the sample's sex, which matched reported sex for all samples (Supplementary Fig. 1b). Furthermore, hierarchical

**Table 1 Characteristics of samples investigated in this study.**

| | | Age (mean ± sd) [years] | Age min/max [years] | PMI (mean ± sd) [hours] | NeuN_pos (mean ± sd)[a] | NeuN_neg (mean ± sd) |
|---|---|---|---|---|---|---|
| DLB | $n = 15$ | 79.6 ± 6.5 | 67–97 | 36.71 ± 19.86 | 0.29 ± 0.06 | 0.73 ± 0.07 |
| Control | $n = 16$ | 81.9 ± 7.95 | 72–92 | 42.08 ± 18.44 | 0.31 ± 0.04 | 0.72 ± 0.03 |
| p-value | | 0.39 | | 0.44 | 0.39 | 0.91 |
| DLB | Males ($n = 8$) | 77.9 ± 6.3 | 69–88 | 35.1 ± 20.3 | 0.3 ± 0.07 | 0.73 ± 0.08 |
| | Females ($n = 7$) | 81.6 ± 9.7 | 67–97 | 38.6 ± 20.8 | 0.29 ± 0.04 | 0.72 ± 0.05 |
| Control | Males ($n = 8$) | 83 ± 6.0 | 76–92 | 44.9 ± 23.6 | 0.31 ± 0.03 | 0.72 ± 0.02 |
| | Females ($n = 8$) | 80.8 ± 7.1 | 72–90 | 39.25 ± 12.5 | 0.31 ± 0.04 | 0.73 ± 0.04 |
| p-value | | 0.564 | | 0.796 | 0.733 | 0.977 |

[a]Estimated neuronal proportion. The p-values reported here refer to the t-test for comparing the mean of DLB and control groups and the one-way ANOVA testing comparing DLB and control groups by sex. sd: standard deviation.

**Fig. 1 The distribution of differentially methylated CpGs. a** QQ-plot of the p-values from the EWAS model. The genomic control inflation rate of the EWAS model is 1.1, indicating a modest inflation. The x-axis shows the expected –log10 (p-value), whereas the y-axis indicates the observed –log10 (p-value). **b** Manhattan plot of the EWAS results between DLB and Control samples. The red line indicates the genome-wide significance threshold (p-value = 9e-8) and blue line represents a q-value of 5 % adjusting for multiple comparisons. The top DMCs per chromosome are annotated.

clustering was performed using the SNP probes to demonstrate genetic similarities among samples and no outliers were observed based on the genetic similarities (Supplementary Fig. 1c).

**Differential methylation site analysis.** We performed differential methylation analysis between the DLB and control samples using the limma package in minfi[33] by correcting for sex, age, PMI, estimated neuronal proportion and the top three PCs inferred from negative control probes. We note that this model returned a genomic control inflation (GCin) of 1.1 (Fig. 1a), indicating a modest inflation. The top signals within each autosome are illustrated in the Manhattan plot (Fig. 1b). We identified 17 differentially methylated sites (DMCs, adjusted p-value < 0.05), which are associated with 23 genes in DLB PM brain. These DMCs showed a mild-to-moderate effect size with the average absolute β-difference of 0.09 (sd = 0.062, range from −0.27 to 0.085, Supplementary Fig. 2a). When following the suggestive genome-wide threshold for EPIC array data[34], we observed only 2 CpGs reaching genome-wide significance at a p-value < 9e-8 (adjusted p-value < 0.02). Following the change-point of QQ-plot in Fig. 1a, we observed 45 CpGs with a p-value < 5.8e-6 (adjusted p-value < 0.1, Supplementary Data 1). These DMCs demonstrate different hypo- and hyper-methylation patterns (Fig. 2a). Moreover, among these 17 DMCs, n = 10 or 59% of the significantly methylated DMCs are hypo-methylated and n = 7 or 41% are hyper-methylated in DLB PM brain. Table 2 lists these 17 DMCs

with the top DMCs being hypo-methylated and the top 2 examples per hypo/hyper DMCs are illustrated in Fig. 2b–e.

We did not observe significant DMCs at a q-value < 0.05 for the known DLB-associated genes (e.g. DLB vs Control across different tissues) as reported in a recent review paper[8]. Most of them were found through low-throughput pyrosequencing or Methylation-specific PCR (e.g. APOE, CRY1, DRD2, FGFR3, PER1, SNCA, and SELENOW)[25–28,35,36] and only few of them were based on genome-wide DNA methylome such as whole-genome bisulfite sequencing (WGBS) or Illumina Array (e.g. ANK1, NBL1, and PTK6)[29,37]. However, we did observe differentially methylated CpGs at p-value < 0.01 for 6 of 10 noted genes (60%) in our study. Specifically, DMCs were detected in the genes ANK1, DRD2, FGFR3, PER1, NBL1, and PTK6.

**Differential methylation region analysis.** To investigate regional differentially methylated signals, we applied DMRCate[38] to identify DMRs. With the seed of DMCs at a q-value < 0.05, we only detected one DMR: upstream of AGPAT1 (chr6:32145376-32146595). To explore more potentially interesting DMRs, we relaxed the FDR q-value thresholds to <0.1 or 0.25. Accordingly, we obtained 2 and 17 DMRs respectively, containing 3 or more consecutive CpGs (ranging between 3 and 33 sites). Of particular interest, the gene AGPAT1 and PAK6 are the two DMRs detected with a FDR q-value < 0.1, representing the hyper-methylated and hypo-methylated DMRs, respectively (see Fig. 2f and Supplementary Fig. 3). Table 3 presents the full list of 17 significantly

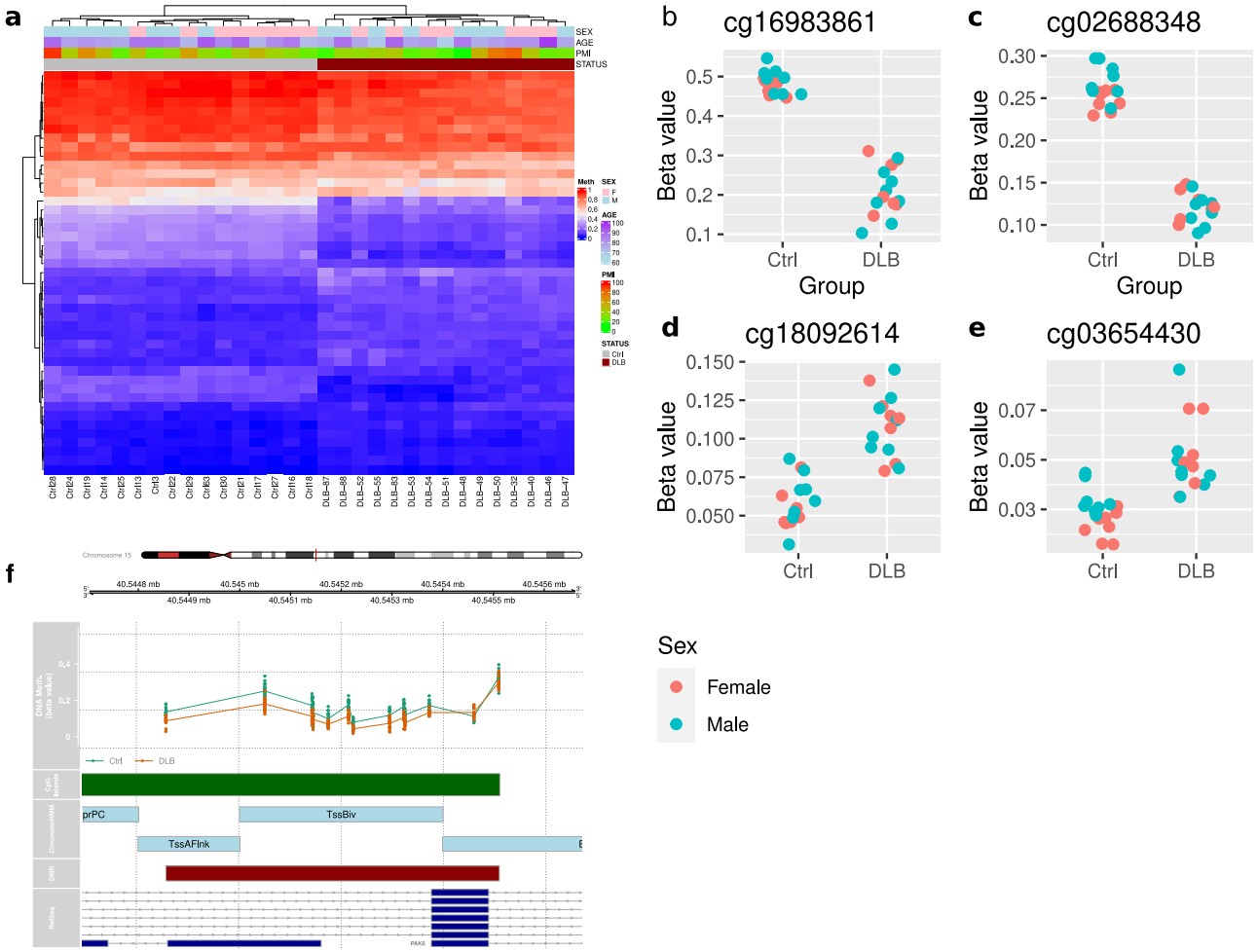

**Fig. 2 DLB-associated DMCs and DMRs. a** The heatmap plot of the significant DMCs with q-value < 0.1 (45 DMCs). The methylation level is scaled by row; blue represents low methylation level and red represents high methylation level. Different phenotype features (including sex, age, PMI and the DLB status) are illustrated in the top plots. **b–e** Scatterplot for the top 2 hypo and hyper DMCs. Samples are colored based on their biological sex. **f** Example of the top DMR (a region associated with the *PAK6*) at q-value < 0.1.

**Table 2 The list of differentially-methylated CpGs (DMCs) at q-value < 0.05.**

| Name | chr | pos | P.Value | adj.P.Val | logFC | Ctrlmean | DLBmean | Gene name |
|---|---|---|---|---|---|---|---|---|
| cg16983861 | chr10 | 91061510 | 2.11E-08 | 1.56E-02 | −2.08 | 0.483 | 0.211 | *IFIT2;LIPA* |
| cg02688348 | chr11 | 57103429 | 7.62E-08 | 1.56E-02 | −1.17 | 0.259 | 0.120 | *SSRP1* |
| cg18092614 | chr17 | 40729792 | 9.43E-08 | 1.56E-02 | 1.67 | 0.058 | 0.109 | *PSMC3IP* |
| cg12360373 | chr16 | 28834065 | 1.00E-07 | 1.56E-02 | −1.73 | 0.148 | 0.050 | *ATXN2L;RP11-1348G14.5* |
| cg21832068 | chr4 | 14889868 | 1.16E-07 | 1.56E-02 | −1.14 | 0.846 | 0.717 | *LINC00504* |
| cg03654430 | chr10 | 91174411 | 1.22E-07 | 1.56E-02 | 1.51 | 0.029 | 0.051 | *LIPA;IFIT5* |
| cg04413320 | chr13 | 58208929 | 2.11E-07 | 1.81E-02 | 1.65 | 0.084 | 0.155 | *PCDH17* |
| cg00153368 | chr11 | 57996423 | 2.12E-07 | 1.81E-02 | −1.00 | 0.911 | 0.853 | *OR10Q1* |
| cg21752471 | chr5 | 133861794 | 2.12E-07 | 1.81E-02 | −1.58 | 0.226 | 0.087 | *PHF15* |
| cg17232014 | chr12 | 13153193 | 3.29E-07 | 2.46E-02 | −1.84 | 0.151 | 0.044 | *HEBP1;HTR7P1;RP11-377D9.3* |
| cg07413941 | chr2 | 149894969 | 3.53E-07 | 2.46E-02 | 1.06 | 0.028 | 0.039 | *LYPD6B* |
| cg09781944 | chr4 | 20254519 | 3.92E-07 | 2.50E-02 | 1.03 | 0.077 | 0.131 | *SLIT2* |
| cg03310027 | chr15 | 43941072 | 6.23E-07 | 3.67E-02 | −1.45 | 0.924 | 0.858 | *CATSPER2;PPIP5K1;CKMT1A;STRC* |
| cg22446331 | chr18 | 19750865 | 8.22E-07 | 4.50E-02 | 0.92 | 0.063 | 0.103 | *GATA6* |
| cg13645001 | chr12 | 22199579 | 9.53E-07 | 4.76E-02 | 1.17 | 0.123 | 0.208 | |
| cg00973947 | chr3 | 143692255 | 9.93E-07 | 4.76E-02 | −1.35 | 0.260 | 0.119 | *C3orf58* |
| cg25882591 | chr17 | 405178 | 1.09E-06 | 4.89E-02 | −0.92 | 0.898 | 0.837 | *RP5-1029F21.3* |

The CpG list is sorted by *p*-values. Name: CpG probe id; chr: chromosome id of the CpG probe; pos: the position of the CpG probe (hg19 version); P.value: the returned two sided *p*-value corresponding to the *t*-statistics from limma; adj.P.val: adjusted *p*-value (or q-value); logFC: log2-fold-change corresponding to the effect (DLB over control); Ctrlmean: average methylation level of control samples; DLBmean: average methylation level of DLB samples; Gene name: the associated gene names according to the complete GENCODE build (version 12).

**Table 3 List of the DMRs detected using DMRCate at q-value < 0.25.**

| chr | start | end | #cpgs | min_fdr | maxdiff | meandiff | gene name |
|---|---|---|---|---|---|---|---|
| chr6 | 32144977 | 32146595 | 33 | 3.68E-18 | 0.159 | 0.068 | *RNF5, AGPAT1* |
| chr15 | 40544719 | 40545794 | 15 | 3.52E-13 | −0.105 | −0.042 | *PAK6, C15orf56* |
| chr10 | 91061114 | 91061844 | 6 | 2.97E-11 | −0.294 | −0.036 | *IFIT2, LIPA* |
| chr16 | 30366434 | 30366960 | 10 | 1.61E-10 | −0.078 | −0.007 | *RP11-347C12.10, CD2BP2* |
| chr17 | 40729397 | 40730125 | 17 | 1.63E-10 | 0.081 | 0.010 | *PSMC3IP* |
| chr20 | 57427145 | 57427942 | 22 | 1.63E-10 | −0.135 | −0.056 | *GNAS* |
| chr10 | 91174098 | 91174667 | 12 | 2.03E-10 | 0.041 | 0.012 | *IFIT5, LIPA* |
| chr14 | 23540583 | 23540892 | 5 | 3.39E-10 | −0.124 | −0.041 | *ACIN1* |
| chr5 | 1801141 | 1801482 | 4 | 1.22E-09 | −0.156 | −0.0495 | *MRPL36* |
| chr15 | 43940866 | 43941072 | 4 | 1.27E-09 | −0.087 | −0.047 | *STRC, CATSPER2* |
| chr6 | 146136370 | 146136749 | 4 | 2.30E-09 | 0.054 | 0.034 | *RP11-545I5.3* |
| chr16 | 857575 | 857981 | 3 | 2.99E-09 | −0.224 | −0.217 | *PRR25* |
| chr2 | 128453107 | 128453484 | 5 | 4.81E-09 | −0.412 | −0.325 | *NA* |
| chr2 | 72079275 | 72079609 | 8 | 5.70E-09 | −0.180 | −0.108 | *NA* |
| chr11 | 57103160 | 57103491 | 7 | 7.63E-09 | −0.114 | −0.011 | *SSRP1* |
| chr21 | 35831870 | 35832008 | 5 | 8.84E-09 | −0.156 | −0.103 | *KCNE1* |
| chr8 | 110661012 | 110661079 | 4 | 1.35E-08 | 0.091 | 0.070 | *SYBU* |

Chr: DMR chromosome id; start: the start coordinate of the DMR (hg19 version); end: the end coordinate of the DMR; #cpgs: the number of CpGs within the DMR; min_fdr: the minimal *p*-value of CpGs within the DMR; Max.diff: maximum of the methylation difference (DLB – control) among CpGs within the DMR; mean.diff: the average of the methylation differences (DLB-control) over CpGs within the DMR. Gene name: the associated genes overlapping DMRs.

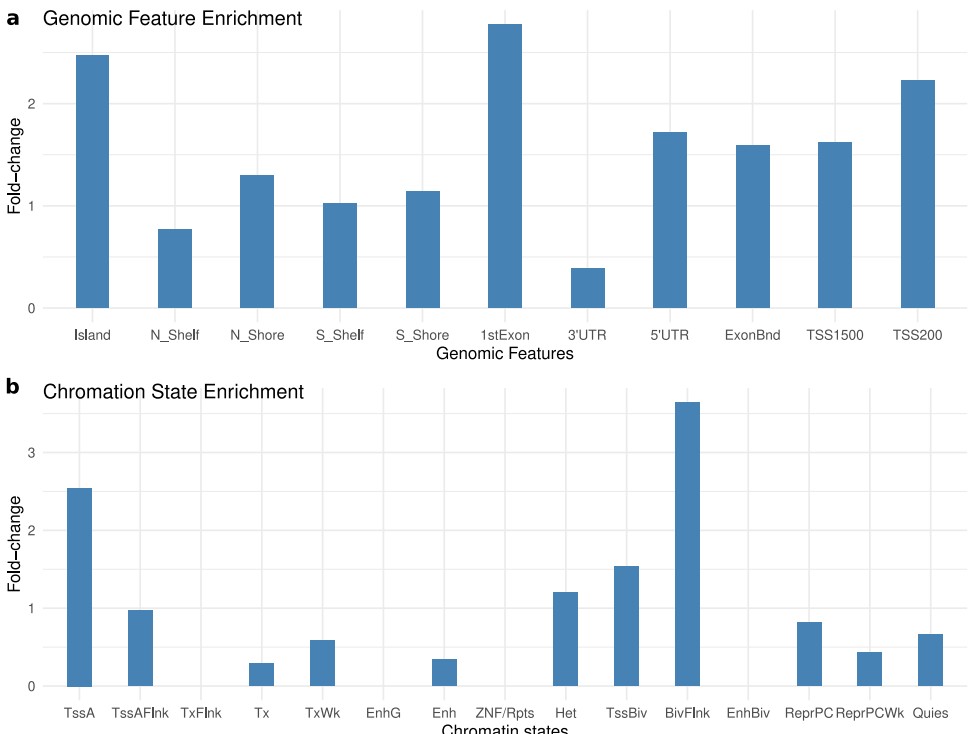

**Fig. 3 Genome feature annotation of DLB-associated DMCs. a** Genomic features and **b** chromatin states enrichment analysis of the DMCs at q-value < 0.25.

methylated DMRs with q-value < 0.25. Like the DMCs, the majority of the DMRs ($n = 12$, 70.1%) were found to be hypo-methylated.

**Genomic feature enrichment analysis on DMCs and DMRs.** As there were only 17 DLB-associated DMCs at q-value < 0.05, it is challenging to perform genomic feature enrichment. Instead, we performed genomic feature enrichment analysis on DMCs with q-value < 0.25, which corresponded to 179 CpGs. We noted that these DMCs showed certain levels of population variation but not

necessarily among the most variable ones (Supplementary Fig. 4). The results of the genomic feature enrichment analysis of DMCs (Fig. 3a) highlights that DMCs are more likely to locate in CpG islands, showing a 2.47-fold significant enrichment ($p$-value $= 1.2 \times 10^{-17}$). These DMCs were also slightly enriched in CpG island shore regions (1.14 and 1.3-fold enrichments for south and north shores, respectively). These DMCs were also enriched in the promoter regions (2.22 and 1.6-fold enrichment for TSS200 and TSS1500, respectively) and first exon (2.7-fold, $p$-value = 9.4e-8). Interestingly, when investigating the enrichment of these DMCs at genomic regions with different chromatin states[39] (E073 brain Dorsolateral Prefrontal Cortex tissue),

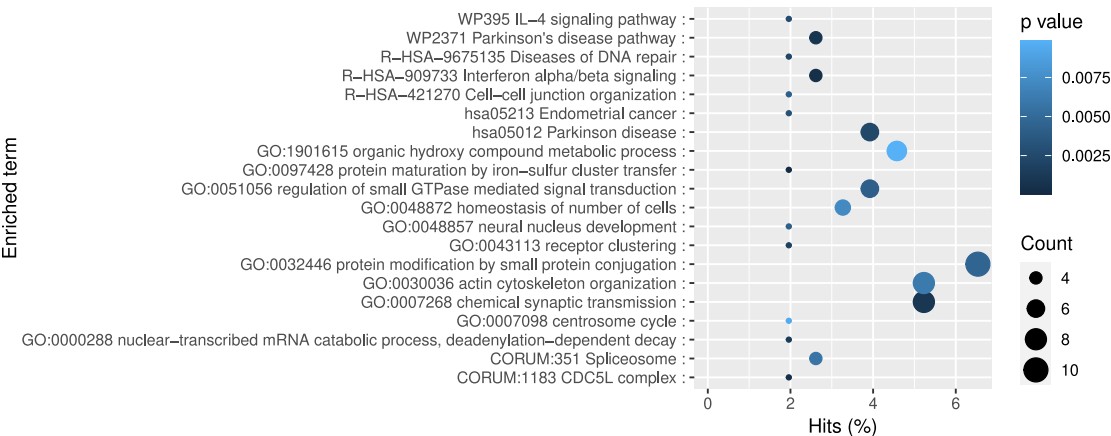

**Fig. 4 Functional enrichment analysis of DLB-associated genes.** Functional enrichment analysis of the DMC-related genes using Metascape. *P* values were indicated with colors while the number of associated genes were indicated with circle sizes.

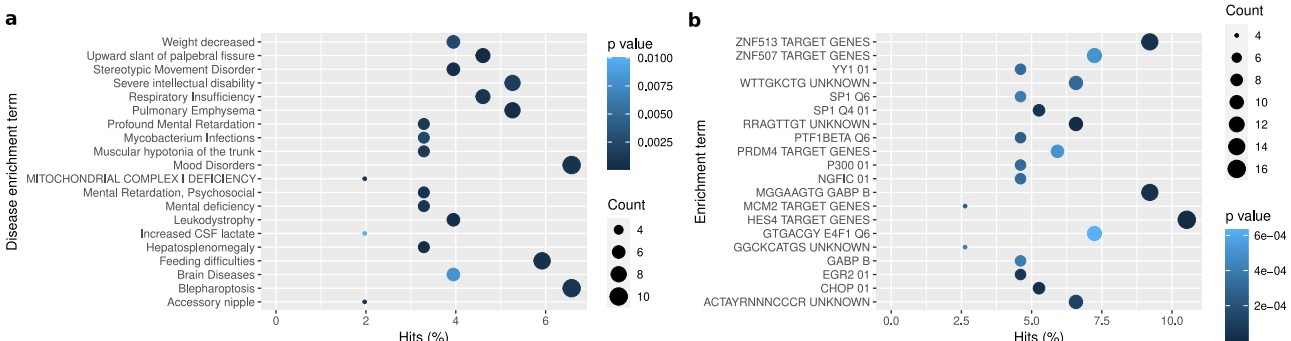

**Fig. 5 Disease and transcription factor targets enrichment analysis of DLB-associated genes. a** DisGeNET analysis for the DMC-associated genes using Metascape. **b** Transcription factor targets enrichment analysis of the DMC-associated genes.

we observed very similar patterns (Fig. 3b). Specifically, these DMCs are enriched in the active promoter region (2.54-fold, *p*-value = 2.2 ×$10^{-4}$), bivalent/poised TSS (1.53-fold) as well as their flanking regions (flanking bivalent TSS) (3.6-fold). Moreover, we reported that they are depleted at quiescent/low state, weak transcription, zinc finger (*ZNF*) genes, repeats regions (satellite repeats that are associated typically with heterochromatin) and genic enhancer regions.

Gene function enrichment analysis using Metascape revealed that the DMCs' associated genes were enriched in the GO biological processes term 'protein maturation by iron-sulfur cluster transfer' (*p*-value = 5.4 × $10^{-5}$), 'chemical synaptic transmission' (*p*-value = 5.4 × $10^{-5}$) and the KEGG pathway 'Interferon alpha/beta signalling' (*p*-value = 4.9 × $10^{-4}$), 'Parkinson's disease pathway' (*p*-value = 8.7 × $10^{-4}$), 'IL-4 signalling pathway' (*p*-value = 2.7 × $10^{-3}$) and 'pathways of neurodegeneration – multiple diseases' (*p*-value = 3.2 × $10^{-3}$) (Fig. 4) as well as 'severe intellectual disability' (*p*-value = 1.58 × $10^{-3}$), and 'brain diseases' (*p*-value = 7.9 × $10^{-3}$) for the DisGeNET analysis (Fig. 5a). It also indicated the enrichment of other neuron development or mental disorders such as 'stereotypic movement disorder' (*p*-value = 3.98 × $10^{-4}$), 'mood disorder' (*p*-value = 7.9 × $10^{-4}$), 'mental retardation, psychosocial' (*p*-value = 7.9 × $10^{-4}$) and 'Mental deficiency' (*p*-value = 1 × $10^{-3}$). These genes also overlap with genes specific to *hRgl3* cells (one of the five radial glia-like cell types detected during human ventral midbrain development)[40] (*p*-value =2.5 × $10^{-3}$). Moreover, they overlap with the binding sites of transcription factor *HES4* (*p*-value = 6.3 × $10^{-8}$), *Ddit3* (*p*-value = 3.2 × $10^{-5}$) and *GABA*

(*p*-value = 3.98 × $10^{-5}$) (Fig. 5b). We note none of the pathways remained significant after FDR *p*-value correction.

To account for the potential biases caused by the differences in the number of CpGs per gene, we performed a gene set enrichment analysis on the DMCs and DMRs associated genes using GOmeth and GOregion functions in the R package missMethyl[41]. Although we used a small numbers of DMCs with a q-value < 0.05, the enrichment analysis returned several neuron development relevant GO terms such as 'corticospinal neuron axon guidance through spinal cord', 'corticospinal neuron axon guidance', and 'cerebral cortex tangential migration using cell-cell interactions' as well as KEGG pathways of 'neuroactive ligand-receptor interaction', 'metabolic pathways', and 'pathways of neurodegeneration-multiple diseases' (Supplementary Fig. 5a). Meanwhile, when we performed the enrichment analysis on genes of DMCs with a q-value<0.25, the 'protein maturation by iron-sulfur cluster transfer' was also observed as the top enriched GO term. Other top enriched GO terms include 'T cell homeostasis' and 'neurotrophin signaling pathway'. Interestingly, these genes were also enriched in the KEGG pathways of 'ubiquitin-mediated proteolysis', 'Ras signaling pathway' and 'prolactin signaling pathway' as well as 'cytosolic DNA-sensing pathway' (Supplementary Fig. 5b). Furthermore, the analysis revealed that DMR-associated genes were enriched in 'regulation of parathyroid hormone secretion', 'regulation of synaptic activity' and highly overlapping with genes involved in 'phospholipase D signaling pathway', 'adrenergic signaling in cardiomyocytes' and 'Parkinson disease' (Supplementary Fig. 5c).

**Weighted gene co-expression network analysis (WGCNA).** To determine if multiple dispersed CpGs can change methylation patterns concordantly yet may remain insignificant ($p > 0.05$) in a site-wise analyses, we applied a clustering analytical approach called weighted gene correlation network analysis (WGCNA). This method is unsupervised and used to identify clusters (modules) of highly correlated features and favours a scale-free network clustering pattern[32]. To characterize DNA methylation changes associated with DLB at the system level, we performed trait correlation against modules identified in the WGCNA analysis (Supplementary Fig. 6). We identified three highly significant module-trait correlations ($|\text{AverageCorr}| > 0.5$, $p$-value $< 8e-4$), with each linked to DLB and Control, and composed of CpGs ranging from 39 to 157 (Supplementary Data 2). All these three modules showed negative correlations with DLB/control states and green-yellow module has the most number of CpGs and significant correlations (MEgreenyellow, $n = 157$, $r = -0.97$, $p$-value $= 2e-19$), tan module in the middle (MEtan, $n = 117$, $r = -0.68$, $p$-value $= 3e-5$) while the midnightblue module has the least number of CpGs (MEmidnightblue, $n = 39$, $r = -0.58$, $p$-value $= 7e-4$) (Supplementary Fig. 7a). Interestingly, we observed that the green-yellow module covered 5 DLB-associated DMCs at q-value $< 0.05$ or 9 DMCs at q-value $< 0.1$. Meanwhile, Module Membership (MM) strongly correlated to Probe Significance (PS) for DLB for each of these 3 modules (MEgreenyellow: $r = 0.98$, $p$-value $< = 1.4e - 110$; MEtan: $r = 0.4$, $p$-value $= 7.9e - 6$); MEmidnightblue: $r = 0.4$, $p$-value $= 0.012$) (Supplementary Fig. 7b–d).

Metascape analysis showed that the genes involved in the green-yellow module were enriched in the biological process of positive regulation of GTPase activity ($p$-value $= 3.5e-5$) and regulation of PID RhoA activity ($p$-value $= 5.2e-5$). By overlapping with disease genes reported in the DisGeNet database, these genes in the green-yellow module were enriched for diseases of Hypoventilation, Lewy Body Disease and neuromuscular diseases (the DisGeNet database, Supplementary Fig. 8a). Furthermore, the enrichment analysis in transcription factor targets indicated that the green-yellow module CpG-associated genes were regulated by TF of *NF1*, *ZNF140* and *DNMT3A* etc. (Supplementary Fig. 8a). The enrichment of muscle system process, regulation of amine transport, fat-soluble vitamin metabolic process and Notch signaling pathway etc. were observed for the tan module (Supplementary Fig. 8b). The enrichment analysis in DisGeNET revealed that the tan module-associated genes were highly overlapped with genes related to diseases of mild cognitive disorder, malignant glioma and memory impairment etc. (Supplementary Fig. 8b).

## Discussion

DLB is a debilitating and difficult-to-diagnose disease. It is well accepted that it is influenced by a combination of genetic and environmental factors[15–18,42]. Epigenetics, reflecting the interplay between genes and environment, has provided the potential for exploring the pathology and etiology of the disease as well as identifying potential biomarkers for diagnosis. In this study, we performed an epigenome-wide association study to identify differentially methylated CpGs in post-mortem brain tissue from DLB cases and control subjects. Comparisons were corrected for sex, age, PMI and also estimated neuronal proportion, which is thought to be heterogeneous across brain tissues. In addition, the model was also corrected for the top variable principal components inferred from negative control probes, which represent potential batch effects. This is one of the first couple of methylome-wide association analysis studies to have been conducted on DLB brain tissue and is comprehensive, using the Illumina EPIC array which covers approximately 850 K CpGs

genome-wide. We detected DMCs and genes that are enriched in the biological process of protein maturation by iron-sulfur cluster transfer, chemical synaptic transmission as well as pathways related to 'Parkinson disease or neurodegeneration – multiple diseases'. We also detected DMC-associated genes overlapped with disease genes for brain diseases, severe intellectual disability as well as mood or mental disorders. This provides independent supportive evidence that aberrant DNA methylation occurs in subjects with DLB pathology, and some of the methylation sites are likely to be epigenetic mechanisms, which may contribute to the development of the disease.

Interestingly, in DLB cases more significant DMCs were observed to be hypo-methylated compared with controls. This may relate to the notable reduction in nuclear levels of Dnmt1[25]. When the α-syn abnormally accumulates in DLB-affected brains, it may cause Dnmt1 to be separated from the nucleus[25]. Our study has catalogued a great many other DMCs and DMRs not previously associated with the disease. Of particular interest, DMR of *PAK6* genomic region was the most hypo-methylated in DLB brain tissue. Notably, *DNMT1* transcriptionally controls *PAK6*, and regulation is required for appropriate cortical interneuron migration[43]. *PAK6* is reported to play a role in the pathogenesis of PD[44]. It can bind to *LRRK2* (leucine-rich repeat kinase 2), a well-known causative gene for PD. The same group also reported that *PAK6* can efficiently phosphorylate *14-3-3γ* at the Ser[59] residue, which can further dissociate the *14-3-3γ* from *LRRK2*[45,46]. Both *LRRK2* and *14-3-3γ* have recently been reported as promising therapeutic targets for age-related neurodegenerative diseases[47,48]. Together, these indicate aberrant *PAK6* methylation sites provide additional information regarding the etiology and pathogenesis of DLB. Meanwhile, the top hyper-methylated DMR was located near the promoter region of both *AGPAT1* and *RNF5*, which have been reported as genes of interest in studies focusing on neurodegenerative diseases such as AD, PD and related dementias[49–52]. Particularly, both *AGPAT1* and *RNF5* were recently observed to be hyper-methylated in AD whole blood samples compared with controls[49] and in PD brain (including dorsal motor nucleus of the vagus and substantia nigra)[53].

Other notable observed DNA methylation changes were *SSRP1*, *PSMC3IP* and *ATXN2L*. Their CpGs are among the most hyper-methylated and hypo-methylated DMCs in DLB brain tissue. *PSMC3IP* was reported to be down-expressed in AD[54] and hyper-methylation of *PSMC3IP* might repress it's gene expression in DLB. It modulates the proteasomal activity and variants within this gene interrupt TF *SIX5* binding, which is associated with AD[54]. *ATXN2L* is a substitute to *ATXN2* and is a member of the spinocerebellar ataxia (SCAs) family, which is associated with a complex group of neurodegenerative disorders[55]. The gain of *ATXN2* function was thought to form the basis for their molecular pathogenesis of neurodegenerative diseases[56]. *SSRP1* is part of the *FACT* complex that is involved in chromatin remodeling during transcription (i.e., it is associated with active RNAPII and promotes nucleosome disassembly to facilitate transcription). This indicates that chromatin remodelling may be involved in the pathology of DLB. The potential role of the interplay between DNA methylation and chromatin remodeling in mediating the disease development may open new lines of research enquiry into mechanisms of DLB pathophysiology. Furthermore, the enrichment analysis highlights DMCs associated genes highly overlapped with the binding sites of transcription factor *HES4* and *CHOP*, where *HES4* is associated with striatal degeneration in post-mortem brain tissue of Huntington's disease[57], and *CHOP* (or alternatively, *DDIT3*), an important regulator of ER stress-induced cell death[58], was reported to be down-regulated in AD[59].

We performed the EWAS analysis and reported the results based on M-values. We also conducted the EWAS analysis based on beta-values. The genomic control inflation lambda for the beta model is 1.06, which is slightly better than the M-value model. Overall, the signals detected by the two models are very similar and the $p$-values are highly correlated (Pearson correlation cor = 0.96) (Supplementary Fig. 9a). In the interests of providing the research community with a more comprehensive resource, we have provided a list of DMCs with a $p$-value < 1e-3 for both models (Supplementary Data 3 & Data 4). Meanwhile, we have also compared the models with and without estimated neuronal cell proportions as co-variates. We observed that the inflation lambda is much larger for the model without estimated neuronal cell proportion correction (i.e. lambda = 1.21) and the correlation of $p$-values between the two models is moderate (cor = 0.61) (Supplementary Fig. 9b). In addition, we compared the models with different numbers of top PCs inferred from the negative control probes. Compared with the models of PCs = 2 or 4, the current model has the lowest genomic control inflation rate (i.e. lambda = 1.11 in current study, comparing with the lambda of 1.16 and 1.18 from the models with top2 PCs or top4 PCs.), although PC2 and PC3 models are more similar compared with the difference between PC4 and PC3 models (cor = 0.98 vs cor = 0.57) (Supplementary Fig. 10).

Recently, Pihlstrøm et al. performed an epigenome-wide association analysis to determine the role of DNA methylation changes in PD and DLB using post-mortem human brain taken from the frontal cortex[30]. They identified 28 DMCs related to Lewy body diseases through a meta-analysis of two independent cohorts while none of them were replicated in our study. Of note, Pihlstrøm et al. used frontal cortex tissue and only the grey matter, whereas our samples were harvested from the parietal cortex, which is responsible for high-order brain functions. In addition, hypo-methylation of *SNCA* (loci at promoter or exon 1 region) has been reported in many studies, both in brain and blood[25,26,60]. However, neither Pihlstrøm et al. nor our study, replicate these reported loci. The different brain tissue regions, different assays and the different disease stages may explain to these discordances. Ideally, we would like to study multiple brain regions in follow-up studies.

In addition to the observations reported herein and the obvious strengths of our study, we do note this study's limitations. Firstly, we do recognize the small sample size used to conduct our study. Specifically, power calculations suggest that we would need 40-50 samples to achieve 80% power for effect sizes ranged between 0.1 and 0.4 which covered the mean effect size and maximal effect size detected in our study (Supplementary Fig. 2b–f). In this way, our sample size in this study is slightly underpowered. Nevertheless, these samples are rare and difficult to obtain: consenting, harvesting, and storing the samples are all very challenging endeavours for DLB parietal cortex. Although we didn't replicate most of the previously reported DLB signals at stringent significance thresholds (q-value < 0.05 or $p$-value < 9e-8), we were able to detect several new signals/genes reported to be associated with other neurodegenerative diseases such as PD and AD. Another limitation of our study is the inability of the standard sodium bisulfite conversion method used in the MethylationEPIC array analysis to distinguish between 5-methylcytosine (5mC) and 5-hydroxymethylcytosine (5hmC), with the latter being enriched in brain cells and thought to play an important role in neural development and neurodegenerative diseases[61,62]. Parallel profiling of methylation and hydroxymethylation[63] or oxidative bisulfite sequencing[64] might be feasible solutions for capturing intermediary DNA methylation states between methylated and unmethylated cytosine.

We acknowledge the limitation due to the design, whereby the DLB and control samples were run on different batches of arrays from Illumina. We used principal components (PCs) constructed from the negative control probes to minimize confounding by batch. Approaches such as surrogate variable analysis, which adjust for the case-control status while calculating the surrogate variables from the majority of probes, led to QQ plots showing substantial inflation of significance. Although we realize that any attempt at correction may eliminate some true signals since the DLB samples and control samples were run on different chips and batches, correcting the model using these PCs (based only on the negative control probes) resulted in better distributional results than other corrections that we explored. We expect this strategy to be fairly conservative since the negative control probes have low fluorescent signals and therefore, this correction may not capture batch effects in high signal ranges. It is worth noting that we previously normalized the methylated and unmethylated signals with funNorm, which uses all types of control probes to align the signals across arrays. After detailed consideration of various options, we decided to implement this relatively conservative correction rather than use a method that will eliminate potential signals of interest. Although we performed functional enrichment analyses on a large set of DMCs with a relaxed significance criteria (i.e. q-value < 0.25, instead of using q-value < 0.01 or < 0.05), the analyses showed certain interesting results which are relevant to DLB and other neurodegenerative diseases. Overall, a larger sample size and meta-analysis with other cohorts would help corroborate the findings reported herein.

In conclusion, we report molecular alterations observed in other neurodegenerative diseases such as PD and AD[29]. The identification of common epigenetic patterns in all these disorders will provide invaluable knowledge about mutual neurodegenerative pathways. Particularly, the observations from comprehensive epigenetic studies of AD and PD might prompt the investigation of similar mechanisms to the study of DLB. Contrastingly, the shared epigenetic and pathological patterns with other types of dementia, with particular reference to PD, make it is more challenging to determine DLB-specific epigenetic regulation. The next logical step therefore would be to directly compare DLB, PD, and AD cases to distinguish between shared and disease-specific changes. Systems-level biology approaches such as genomics, epigenomics, transcriptomics, proteomics and metabolomics are improving our understanding of the complex interplay between molecules in disease pathogenesis[65]. For instance, the top CpG detected in our study was located within the *LIPA* gene and the promoter of *IFIT2*. *LIPA* is an enzyme whose oxysterol products activate *APP*[66], which is implicated in AD[67]. Meanwhile, our group have previously provided some insight into the changes in the brain metabolome of DLB[68,69]. Therefore, another major challenge in comprehensively understanding the onset of DLB and its rapid progression, is the integration of a number of 'omic' platforms to provide a 'global' biochemical profile. This should pave the way for the development of drug targets and prospective biomarker panels for early detection, and potentially a means of predicting those at greatest risk of developing DLB.

## Methods

**Study cohort**. Post-mortem brain tissue samples (harvested from the parietal cortex, Brodmann area 7) were obtained from pathologist confirmed DLB cases ($n = 15$) and non-cognitively impaired controls ($n = 16$). These samples were selected to ensure that groups were sex and age matched. Tissues were obtained by the application from the Brains for Dementia Research (BDR) Initiative[70,71], Institute of Clinical Neurosciences, School of Clinical Sciences, University of Bristol, Bristol, UK. We evaluated DLB post-mortem brain based all medical history on archival information (albeit limited) and neuropathology reports. Control brains were selected from subjects who had no clinical history of dementia, few or

no neuritic plaques, a Braak tangle stage of III or less and no other significant neuropathological abnormalities. The details of the samples could be found in Supplementary Table 1. This study was approved by the Beaumont Health System's Human Investigation Committee (HIC No.: 2018-387).

**DNA methylation profiling**. DNA was extracted from the lyophilized and milled brain tissue using QIAamp DNA Mini Kit (Qiagen) and the DNA samples were bisulfite converted using EZ DNA Methylation-Direct Kit (Zymo Research, Orange, CA) according to manufacturer's protocol. We used 500 ng of DNA to perform bisulfilte conversion. Followed by this, Illumina Infinium MethylatioNEPIC BeadChip arrays (or EPIC array) with >850,000 cytosine sites per assay were used to profile DNA methylation. We followed the Illumina protocol to perform the assay followed by BeadChip imaging using Illumina iScan (Illumina, California, USA).

**Statistics and reproducibility**
*DNA methylation analysis*. The Illumina EPIC array raw intensity data were processed and analyzed using the methylation analysis R package Minfi (version 1.36.0)[33,72]. Specifically, the background was subtracted/adjusted using the "noob" method[73]. Multiple filters were applied for the pre-processing. Probes were removed when any bases in their target sequence overlapped with known single nucleotide polymorphisms (SNPs), or overlapped with non-CpG context (i.e., CpA, CpC and CpT); probes with unreliable measurements (detection $p$-value > 0.01) in any of the samples along with those probes previously described to hybridize to multiple locations in the genome[74] were also removed. In addition, probes located on X and Y chromosomes were also excluded. Furthermore, signal intensities for both methylated and unmethylated channels were calculated and compared to check for bad-quality samples. Sex chromosome CpGs were used to estimate sample sex with getSex() function in minfi and SNPs (based on RS probes, i.e. 59 explicit SNP probes on the Illumina EPIC array) profiles were extracted to perform a hierarchical clustering to check for any mix-up in samples. To minimize technical variation in signal intensities, DNA methylation intensity profiles were further normalized using the functional normalization approach[75] – i.e. the pre-processFunnorm in the minfi R package.

Methylation level for the remaining probes were measured using the standard beta value, which represents the ratio of the methylated signal intensity to the sum of both methylated and unmethylated signals. The beta value ranged from 0 to 1 with 0 being completely unmethylated and 1 being completely methylated. Meanwhile, the M-value (or logit(beta)) was obtained for the differential methylation analysis[76]. Specifically, the differential methylation analysis between DLBs and controls were performed using linear regression modeling, implemented in limma (version 3.46.0)[77], on each CpG adjusting for age, sex, post-mortem interval (PMI), estimated neuronal proportions (i.e., the neuronal fraction and non-neuronal fraction), and inferred top principal components (PCs) from the negative control probes. There are mainly only two primary cells types – neuronal and glial in the brain tissue we studied. The neuronal proportions were estimated using R package estimateCellCounts with the setting for compositeCellType=DLPFC and with the reference data FlowSorted.DLPFC.450k (version 1.26.0)[72]. The principal component analysis (PCA) analysis was performed on the negative control probes ($n = 411$) to estimate any potential batch effects. The top three PCs were selected by checking the scree plot (Supplementary Fig. 11) and the first two PCs showed significant associations with EPIC chip slides by the Kruskal-Wallis rank sum test ($p$-value = 0.004 and 0.0003, respectively). The $P$ values from the empirical Bayes moderated t-statistics in the limma model were further calculated. Results of the limma models are summarized in QQ-plots and Manhattan plots. To correct for multiple comparisons, Benjamini and Hochberg false discovery rate (FDR) correction was used. DMCs were identified as adjusted $p$-value < 0.05. To compare the influences of different co-variates on fitting the model, we have tried models without neuronal proportion correction, and with different number of PCs. The genomic control inflation (GCin, or lambda) and the correlation of p-values between different models were used to compare the models. In addition, differentially methylated regions (DMRs) were analyzed using R package DMRcate (version 2.4.1)[38] with the same covariate corrections and default settings. Lastly, we adopted the R package "pwrEWAS"[78], which was designed to estimate the power specific to EWAS studies, to perform the power analysis. It provided the power estimation for the most commonly used tissue types for EWAS but no brain tissue. We thus explored multiple tissues included in the package with hope to have an approximate estimation for brain. To mimic the scenario we have in this study, we selected balanced sample size per group (i.e., sample rate for group 1 is 0.5), the number of CpGs to be tested as 800,000, multiple testing correction FDR threshold as 0.05 and target number of DMCs as 30. Since we do not know the likely range of effect sizes for the DLB-associated DMCs, we have calculated power using four values for the mean methylation difference between two groups: 0.1, 0.2, 0.3 and 0.4.

*Enrichment analysis*. The annotation information for genome features including gene structures (such as transcription start site, TSS200 and TSS1500; exon; first exon; intron; 5'UTR; and 3'UTR) as well as CpG island structures were extracted directly from the associated annotation table provided with the EPIC assay.

The chromatin states inferred using ChromHMM[39] were downloaded from Roadmap Epigenomics (https://egg2.wustl.edu/roadmap/web_portal/chr_state_learning.html) where the 15 states model for prefrontal cortex was selected (E073).

Fold-changes were calculated based on the ratio of DMCs in each gene feature/regulatory element against the ratio of background CpGs in each gene feature/regulatory element. Fisher exact test was used to calculate the enrichment significance and a $p$-value < 0.05 was considered as statistically significant. We noted that we have used the chromatin states inferred from the prefrontal cortex of the brain rather than the parietal cortex, since chromatin state inferences for the latter are not currently available from the RoadMap Epigenomic Project. Thus, the enrichment results should be treated with a certain level of caution and considered suggestions rather than accurate predictions. However, given the physical proximity of the prefrontal cortex to the parietal cortex, we believe the enrichment analysis provides results of biological relevance regarding the regulatory regions.

In order to determine the biological significance of DLB-related DMCs or DMRs, gene ontology, pathway enrichment, disease-associated gene enrichment and transcription regulatory network as well as the transcription factor binding site enrichment analysis were performed using Metascape[79]. In addition, in order to account for the potential biases caused by differences in the number of CpG sites per gene, we also used R package missMethyl (version 1.24.0)[41] to perform the GO and KEGG enrichment analysis.

**Weighted gene-co-expression network analysis (WGCNA)**. Weighted gene co-expression network analysis (WGCNA)[31,32], aimed at detecting highly correlated modules of CpGs, was done using the top 5% most variable CpGs ($n = 38320$). The WGCNA was performed using the WGCNA package in R[32] with the default parameters. A soft-thresholding power of 14, for which the scale-free topology fit index curve flattens out upon reaching a high value of signed R-square of 0.8, was chosen. A maximum block size of 40,000 and a signed network with biweight mid-correlation were used (to construct a topological overlap matrix) and a minimum number of CpGs per module = 30 and cut height = 0.2 were used to detect methylation modules. The module eigengene (ME) value was calculated for each module and further the Pearson correlation between MEs and traits was computed to quantify module-trait associations.

**Reporting summary**. Further information on research design is available in the Nature Research Reporting Summary linked to this article.

## Data availability
The raw data of DNA methylation profiles generated in this study were submitted to Gene Expression Omnibus (GEO): GSE190348. Source data underlying Figs.1–5 are presented in Supplementary Data 5–7.

## Code availability
This study does not use any custom code or mathematical algorithm. All the relevant R code or software used in this study has been detailed in the Methods section. In addition, the analysis code is available at https://github.com/xshaonrc/DLBepigenetics.

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

## Acknowledgments

X.S. and M. C. are supported by the National Research Council Canada through the Aging in Place Challenge Program. This work was partly funded by the generous contribution made by the John and Marilyn Bishop Charitable Foundation and the Fred A. & Barbara M. Erb Foundation. S.F.G.'s laboratory is supported by grants from the National Institute of Neurological Disorders and Stroke (1R01NS110838- 01A1), the National Institute on Aging (1R21AG067083-01) and the Michael J. Fox Foundation (MJFF16201). B.D.G.'s laboratory has received support for AD research from Alzheimer's Research UK (ARUK-NC2019-NI), the Medical Research Council (MRC) (CIC-CD1718-CIC25), US-Ireland Health and Social Care NI (HSC R&DST/5460/2018) and InvestNI (RD101427 11-01-17-008).

## Author contributions

Conceptualization, X.S., S.V., M.C. and S.F.G.; Methodology, X.S., S.V., A.Y., C.M.T.G., A.S., U.R., S.B., M.C. and S.F.G.; Formal Analysis, X.S., S.V and M.C.; Supervision, S.F.G.; Writing—Original Draft Preparation, X.S.; Writing—Review and Editing, X.S., S.V., C.M.T.G., B.D.G., P.P., P.G.K., B.M., M.C. M.M. and S.F.G. All authors have read and agreed to the published version of the manuscript.

## Competing interests

S.F.G. receives commercial support as a consultant from Biogen, Coleman Research and Roche which is not related to the results of this study. No other authors have competing interests.

## Informed consent statement

Informed consent was obtained from all subjects involved in the study.

## Institutional review board statement

The study was conducted according to the guidelines of the Declaration of Helsinki. This study was approved by the Beaumont Health System's Human Investigation Committee (HIC No.: 2018-387).
