## [Peer Review File · Communications Biology]

Reviewers' comments:

Reviewer #1 (Remarks to the Author):

The work looks at the DNA methylation profile of 31 post mortem brain samples from Dementia with Lewy bodies (DLB) patients and controls using the Illumina EPIC array. Overall this is a well written manuscript with some interesting points in a new area as very little work on the epigenetics of DLB has been published to date. However, I do have some fundamental issues with the results that means at this stage I cannot recommend publication.

- Does the manuscript have technical or conceptual flaws that should prohibit its publication? If so, please provide details.

This study has a small sample size for an epigenome wide association study. Despite this, the authors report detection of 296 differentially methylated sites and 215 differentially methylated regions (DMRs), this is far in excess of what would be expected for a sample cohort of this size. Potential reasons for this could be:

- o A lack of sample randomisation in the running of the EPIC arrays, the manuscript does not state randomisation occurred. This would cause huge batch effects that would not be correctable through the use of surrogate variables (SVs).

- o Inflation lambda statistics should be included to show that the findings of this study are not over inflated.

- o Detail on the criteria of DLB pathology for sample inclusion is missing from the manuscript. From post-mortem tissue only it is not possible to determine DLB from Parkinson's disease. How are the authors confident that this doesn't bring variation into their results?

- o Similarly the criteria for control sample inclusion is equally vague. The authors state the use of "non-cognitively impaired controls". How was cognitive ability assessed and were any pathology levels measured in control tissue?

- o Brain region studied seems inconsistent, originally stated as Neocortex, Brodmann area 9, but later stated as only Parietal Cortex. As the Neocortex is a large area of the human brain caution is needed in interpreting results as this would introduce brain region variation in the methylation data. Could the authors please clarify that all samples were from the same brain region?

In addition fundamental QC metrics are missing from the methods including:

- o Signal intensity check of both methylated and unmethylated channels.

- o Sex check and hierarchical clustering methods (based on RS probes) to be sure of no sample mix ups at the array stage.

- o There was also no note in the methods listing how many samples/ probes were lost due to p filtering, cross-hybridising, SNP probe removal etc.

Following the methylation data analysis the authors make use of ChromHMM data from Roadmap Epigenomics. This data was from prefrontal cortex only and not neocortex or parietal cortex as used in this study. Therefore any conclusions drawn from the interaction of DNA methylation and chromatin state is likely overstated.

There were few grammatical and spelling errors in the manuscript and all tables, but particularly table 2 would benefit from a clear legend.

- Are the conclusions original? If not, please provide relevant references.

I find the conclusions to be original and interesting take on the field however given the highly questionable significance of the data presented the conclusions are likely over stated.

- Do you feel that the results presented are of immediate relevance for people in your own discipline or for a broader audience? If you recommend publication, please outline briefly what you consider to be the outstanding features.

The results of this study would be of interest to the DLB field generally however in its current state I cannot recommend publication.

- If you feel that specific additional experiments would strengthen the case for publication in Communications Biology, please provide suggestions.

I would strongly recommend a re-analysis of this data to determine if the findings are real and not a result of batch effects or other factors in the experimental design. It would also be of benefit to replicate these findings in a targeted manner using an alternative technology, for example bisulfite pyrosequencing.

Reviewer #2 (Remarks to the Author):

The authors of this study are examining methylation indices in cases of dementia with Lewy bodies. This is an unmet need in the field and influenced from underwhelming progress of genome wide association studies. The study results are an important step forward towards understanding the pathogenesis of dementia that in actuality - represent an oversimplified understanding of the influence of genomic methylation and its relevance towards biomarker validation.

Main points.

Abstract: this is poorly described with the reader having to guess what type of technology used to distinguish DNA methylation. Furthermore, the number of DLB cases (15) and NCI controls (16) are not described. Likewise, adding effect sizes and statistical significance to findings would be helpful.

Statistical considerations: Study size is without question small and difficult to determine how broad the methylation signatures identified would replicate in larger DLB populations. Is the study too underpowered and what are the power calculations?

Have the authors accounted for methylation differences that are known to be associated with age and sex? What other clinical variables exist for the cohort, such as smoking, medication, etc? These details if available should be included in the study for readers to assess.

What is the significance of reduced DLB methylation? Do the authors believe hypomethylation converge on transcription factor binding sites (TFBS) to regulate gene expression changes? Are TFBS represented in the WGCNA? Are the authors proposing reduced methylation with DLB pathogenesis?

The authors should specifically discuss the study limitations clearly.

Cell type considerations: the influence of cell-type specific DNA methylation is briefly described. The authors should be considering analytical methods like Cell DMC or equivalents and discussing results with/without cell type adjustment. What is the impact of different cell type influences such as estimateCellCounts2 or methylCIBERSORT?

While the methylation data is interesting the above questions should be considered in light of dementia pathogenesis and is relevant to biomarker validation.

Reviewer #3 (Remarks to the Author):

This study represents a thorough assessment of differences in DNA methylation Brodmann area 7 of the post-mortem brains of patients with Dementia with Lewy Bodies (DPB) and controls. Several differentially methylated sites and regions are identified, with several of these sites and regions identifying genes that are interesting and plausible candidates for DLB. The authors make the interesting observation that there is a bias towards identifying hypomethylation in the cases and they link this to a previously observed reduction in nuclear Dnmt1 in the post-mortem brains of patients with DLB and a mouse model of the condition. Further follow-up analyses to explore this bias would be of interest and could be performed by tweaking the existing functional and genomic enrichment analyses. Weighted gene coexpression network is performed and represents a useful and informative extension to the other enrichment analyses that are performed. These findings are interesting and novel, being one of the first post-mortem explorations of DLB. The main limitation of the study is the relatively small sample size. I do not feel this is adequately addressed either in terms of presenting power calculations to attempt to identify the magnitude of

the methylation differences that are possible to identify in this sample or simply acknowledging the effects of the small sample in the discussion. Since this is an early study of post-mortem DNA methylation in DLB and the first to study this particular brain region, I think the results are likely to be of broad interest but the limitations of the sample size must be fully addressed.

The manuscript is fairly clearly written, although some grammatical errors (e.g. missing commas) hamper its readability in places. Some additional details are required in the methods section to permit reproduction of the analyses (I note these below). In general, the statistical analyses appear sound, although I have some comments below about the unit of methylation measurement used for analyses and the choice of significance threshold and claims made regarding the FDR correction and the small sample size. The conclusion would benefit from some re-writing to make sure the main messages and limitations of the study are clearly and concisely presented.

I have made some specific comments below:

Major comments:

1. "Such mechanisms regulate the transcriptomes of neuronal cells, and play important roles in neurodegeneration.". Please provide citations to support these claims. It would also be good to provide more specific information about DNA methylation, since this is the focus of the study.
2. "These studies are extremely limited because they are either based on targeted candidate genes or were statistically underpowered". Please can you elaborate on why you think they were underpowered? Can you also set out in the description (in the next paragraph) of your study why it is adequately powered/better powered than previous efforts? The sample sizes studied in this study are small for an EWAS and small compared to those studied by a recent study of DNA methylation in DLB by Pihlstrom et al. The issue of the sample size in this study needs to be addressed more adequately. What size effects were you powered to detect? Although Pihlstrom et al. studied a different brain region, it may be useful to perform a look-up of the significant CpGs identified in this study to look for replication in Pihlstrom et al.
3. What was the rationale for analysing beta-values rather than M-values, which have statistically preferable properties (see Du et al., 2010)?
4. The justification for using an FDR correction and not one of the recommended EWAS significance thresholds as the primary approach is unclear. It is not necessary to use an FDR correction in order to provide a resource to researchers as the authors claim, as the same could be achieved using a standard EWAS threshold and providing full summary statistics for all tested CpGs. It is also incorrect to describe the FDR correction as "extremely stringent" or as a compensation for the small sample sizes. I can understand why it might be preferable to perform functional/genomic enrichment analyses using a list of CpG sites determined through a more relaxed significance threshold but a standard EWAS significance threshold (either Saffari et al., 2018 or Mansell et al., 2019) should represent the primary approach for declaring significance. In the discussion, reference is made to the fact that previous studies have applied p-values that do not control for effect size. It is unclear what this means or how this study differs in its approach. Moreover, it isn't clear (given limited understanding of what differences in DNA methylation mean) whether controlling for effect size would even be desirable.
5. Please include a Q-Q plot for the EWAS in the supplementary materials and comment on it in the results section.
6. The enrichment analysis provides interesting insights into the differentially methylated CpGs. It is possible, however, that enrichment in particular genomic locations or chromatin states is seen simply because these locations are enriched for CpG sites that show variation in a population (and, by definition, the associated CpGs must show variation). That is, since many CpG sites on the array are non-variable, perhaps by using all CpGs on the array as the background set, one simply sees enrichment in areas of the genome/chromatin states that contain variably methylated sites. It might be possible to test this idea by (i) assessing how the sites that are associated with DLB rank in terms of variability compared to the whole array and (ii) selecting a background set that is matched for variability.
7. What happens if the enrichment analysis is performed for hyper- and hypo-methylated sites separately?
8. Did your approach to GO analysis account for biases caused by differences in the number of CpG sites per gene (Maksimovic et al., 2021)? If not, consider using the KEGG and GO tools in missMethyl where it is possible to apply this correction. When reporting the results from the functional enrichment analysis, it needs to be clearer upfront that no associations survived FDR correction (and also how far the best p-value is from being significant after FDR adjustment).
9. "This provides clear evidence that aberrant DNA methylation occurs in subjects with DLB

pathology, and some of the methylation sites are likely to be epigenetic mechanisms which contribute to the development of the disease." It is not possible to assert causality from the results presented in this manuscript, this claim should be modified or further justified.

10. The approach of using an FDR adjustment to correct for multiple testing cannot be described as "extremely stringent"

Minor comments:

1. In the abstract, state which area of the brain was studied.

2. Make it clear that SNCA is the gene encoding alpha synuclein-you do this later in the introduction but perhaps move the explanation to the first mention of the gene name.

3. "toxin like" should be "toxin-like"

4. "Epigenetic mechanisms including the methylation of CpG dinucleotides on genomic DNA, mediate the gene-environment interactions thought to be involved in the development of the disease"

This sentence is ambiguous: it can either be read as a claim that epigenetic mechanisms mediate the gene-environment interactions that are believed to be involved in DLB (which is quite a strong claim that would definitely need a supporting reference) or that epigenetic mechanisms mediate gene-environment interactions in general, which is, of course, a widely accepted belief. Please make it clear which meaning is correct.

5. "Compared with other neurodegenerative diseases such as PD and AD, where comprehensive studies of DNA methylation regulation have been conducted, very few studies have undertaken this for DLB"

It is true that comprehensive studies of DNA methylation have been carried out for these conditions; however, these studies have generally just identified sites showing differences in methylation rather than showing that the DNA methylation is actively involved in regulating anything. As such, the term "DNA methylation regulation" is misleading. In many cases, the role of DNA methylation is unknown and, in many places in the genome, variation in methylation does not appear to be correlated with gene expression.

6. "Desplats et al. (2011) found global DNA hypo-methylation...": clarify whether this is the SNCA gene or the whole genome.

7. "ANK1, a well-known epigenetic target of AD": not accurate to call ANK1 a target of AD as this suggests a direction of effect that is not known.

8. "Gender" should be "sex"

9. "Probes when the last 3 bases in its target sequence overlapped with known single nucleotide polymorphisms (SNPs)": did you only consider SNPs with minor allele frequency above a certain threshold? Is there a study that supports the use of the three base cut-off? I am aware of the recommendation to use a five base cut-off by Zhou et al. (2017). This sentence also contradicts the first sentence in the "Differential methylation site analysis" section of the results section, where you state "Following the removal of all probes which overlapped with known single nucleotide polymorphisms". Please clarify the approach used.

10. "overlapped with non-CpG context": can you clarify what this means?

11. "probes with unreliable measurements (p-value>0.05)": clarify that this is the detection p-value

12. "along with those probes previously described to hybridize to multiple locations in the genome were removed". Please cite the reference you used to identify these probes.

13. Table 1: add the unit of measurement for post-mortem interval

14. "These DMCs showed a mild-to-moderate effect size with the average absolute β -difference of 0.07 (sd= 0.021, range from -0.18 to 0.16)". I am not convinced that this is a useful statement. I don't think an average beta difference is a particularly useful figure to report, particularly when the causes and consequences of variation in DNA methylation are poorly understood and appear to vary according to context (i.e. a small difference might be very important in some contexts whilst a large difference might have little effect in others). It's also worth noting that that effect of a given change in beta-value will quite possibly vary depending on where on the scale of possible beta-values this change occurs (i.e. where between 0 and 1).

15. It would be preferable for the figures in Figure 1A to be slightly larger

16. Table 2 should be ordered on p-value as there are ties between the FDR adjusted p-values. Gene names should be italicised (in other tables too).

17. Some words are unnecessarily capitalised (bivalent, zinc)

18. Please clarify the meaning of this: "...the majority were not captured by marginal DLB associations and may represent important, additional loci for DLB related epigenetic changes"

19. Use "methylome-wide association study" or "epigenome-wide association study" rather than "genome-wide association study"
20. The meaning of "intelligence related diseases" is not clear
21. "Our study corroborates some previously reported genes that were associated with DLB". Explain how these genes have been implicated in DBP (i.e. GWAS or EWAS etc)
22. The findings re. PAK6 are interesting: is there any evidence linking altered methylation at the associated sites to an effect on expression of PAK6?
23. "To the best of our knowledge, this is the first reported DNA methylation association analysis to have been conducted on DLB brain tissue". How does this relate to the study of Pihlstrøm et al., which is also discussed?
24. The bias towards finding hypomethylation in the DLB cases is an interesting finding. How does this relate to the findings of Pihlstrøm et al.? Are there any studies of gene expression in DLB (preferably brain) that also show a bias in the direction of the associated gene expression changes?

Reviewers' comments:

Reviewer #1 (Remarks to the Author):

The work looks at the DNA methylation profile of 31 post mortem brain samples from Dementia with Lewy bodies (DLB) patients and controls using the Illumina EPIC array. Overall this is a well written manuscript with some interesting points in a new area as very little work on the epigenetics of DLB has been published to date. However, I do have some fundamental issues with the results that means at this stage I cannot recommend publication.

- Does the manuscript have technical or conceptual flaws that should prohibit its publication? If so, please provide details.

This study has a small sample size for an epigenome wide association study. Despite this, the authors report detection of 296 differentially methylated sites and 215 differentially methylated regions (DMRs), this is far in excess of what would be expected for a sample cohort of this size. Potential reasons for this could be:

- o A lack of sample randomisation in the running of the EPIC arrays, the manuscript does not state randomisation occurred. This would cause huge batch effects that would not be correctable through the use of surrogate variables (SVs).

- o Inflation lambda statistics should be included to show that the findings of this study are not over inflated.

- o Detail on the criteria of DLB pathology for sample inclusion is missing from the manuscript. From post-mortem tissue only it is not possible to determine DLB from Parkinson's disease. How are the authors confident that this doesn't bring variation into their results?

- o Similarly the criteria for control sample inclusion is equally vague. The authors state the use of "non-cognitively impaired controls". How was cognitive ability assessed and were any pathology levels measured in control tissue?

- o Brain region studied seems inconsistent, originally stated as Neocortex, Brodmann area 9, but later stated as only Parietal Cortex. As the Neocortex is a large area of the human brain caution is needed in interpreting results as this would introduce brain region variation in the methylation data. Could the authors please clarify that all samples were from the same brain region?

In addition fundamental QC metrics are missing from the methods including:

- o Signal intensity check of both methylated and unmethylated channels.

- o Sex check and hierarchical clustering methods (based on RS probes) to be sure of no sample mix ups at the array stage.

- o There was also no note in the methods listing how many samples/ probes were lost due to p filtering, cross-hybridising, SNP probe removal etc.

Following the methylation data analysis the authors make use of ChromHMM data from Roadmap Epigenomics. This data was from prefrontal cortex only and not neocortex or parietal cortex as used in this study. Therefore any conclusions drawn from the interaction of DNA methylation and chromatin state is likely overstated.

There were few grammatical and spelling errors in the manuscript and all tables, but particularly table 2 would benefit from a clear legend.

- Are the conclusions original? If not, please provide relevant references.

I find the conclusions to be original and interesting take on the field however given the highly questionable significance of the data presented the conclusions are likely over stated.

- Do you feel that the results presented are of immediate relevance for people in your own discipline or for a broader audience? If you recommend publication, please outline briefly what you consider to be the outstanding features.

The results of this study would be of interest to the DLB field generally however in its current state I cannot recommend publication.

- If you feel that specific additional experiments would strengthen the case for publication in Communications Biology, please provide suggestions.

I would strongly recommend a re-analysis of this data to determine if the findings are real and not a result of batch effects or other factors in the experimental design. It would also be of benefit to replicate these findings in a targeted manner using an alternative technology, for example bisulfite pyrosequencing.

Reviewer #2 (Remarks to the Author):

The authors of this study are examining methylation indices in cases of dementia with Lewy bodies. This is an unmet need in the field and influenced from underwhelming progress of genome wide association studies. The study results are an important step forward towards understanding the pathogenesis of dementia that in actuality - represent an oversimplified understanding of the influence of genomic methylation and its relevance towards biomarker validation.

Main points.

Abstract: this is poorly described with the reader having to guess what type of technology used to distinguish DNA methylation. Furthermore, the number of DLB cases (15) and NCI controls (16) are not described. Likewise, adding effect sizes and statistical significance to findings would be helpful.

Statistical considerations: Study size is without question small and difficult to determine how broad the methylation signatures identified would replicate in larger DLB populations. Is the study too underpowered and what are the power calculations?

Have the authors accounted for methylation differences that are known to be associated with age and sex? What other clinical variables exist for the cohort, such as smoking, medication, etc? These details if available should be included in the study for readers to assess.

What is the significance of reduced DLB methylation? Do the authors believe hypomethylation converge on transcription factor binding sites (TFBS) to regulate gene expression changes? Are TFBS represented

in the WGCNA? Are the authors proposing reduced methylation with DLB pathogenesis?

The authors should specifically discuss the study limitations clearly.

Cell type considerations: the influence of cell-type specific DNA methylation is briefly described. The authors should be considering analytical methods like Cell DMC or equivalents and discussing results with/without cell type adjustment. What is the impact of different cell type influences such as estimateCellCounts2 or methylCIBERSORT?

While the methylation data is interesting the above questions should be considered in light of dementia pathogenesis and is relevant to biomarker validation.

Reviewer #3 (Remarks to the Author):

This study represents a thorough assessment of differences in DNA methylation Brodmann area 7 of the post-mortem brains of patients with Dementia with Lewy Bodies (DPB) and controls. Several differentially methylated sites and regions are identified, with several of these sites and regions identifying genes that are interesting and plausible candidates for DLB. The authors make the interesting observation that there is a bias towards identifying hypomethylation in the cases and they link this to a previously observed reduction in nuclear Dnmt1 in the post-mortem brains of patients with DLB and a mouse model of the condition. Further follow-up analyses to explore this bias would be of interest and could be performed by tweaking the existing functional and genomic enrichment analyses. Weighted gene coexpression network is performed and represents a useful and informative extension to the other enrichment analyses that are performed.

These findings are interesting and novel, being one of the first post-mortem explorations of DLB. The main limitation of the study is the relatively small sample size. I do not feel this is adequately addressed either in terms of presenting power calculations to attempt to identify the magnitude of the methylation differences that are possible to identify in this sample or simply acknowledging the effects of the small sample in the discussion. Since this is an early study of post-mortem DNA methylation in DLB and the first to study this particular brain region, I think the results are likely to be of broad interest but the limitations of the sample size must be fully addressed.

The manuscript is fairly clearly written, although some grammatical errors (e.g. missing commas) hamper its readability in places. Some additional details are required in the methods section to permit reproduction of the analyses (I note these below). In general, the statistical analyses appear sound, although I have some comments below about the unit of methylation measurement used for analyses and the choice of significance threshold and claims made regarding the FDR correction and the small sample size. The conclusion would benefit from some re-writing to make sure the main messages and limitations of the study are clearly and concisely presented.

I have made some specific comments below:

Major comments:

1. "Such mechanisms regulate the transcriptomes of neuronal cells, and play important roles in neurodegeneration." Please provide citations to support these claims. It would also be good to provide more specific information about DNA methylation, since this is the focus of the study.

2. “These studies are extremely limited because they are either based on targeted candidate genes or were statistically underpowered”. Please can you elaborate on why you think they were underpowered? Can you also set out in the description (in the next paragraph) of your study why it is adequately powered/better powered than previous efforts? The sample sizes studied in this study are small for an EWAS and small compared to those studied by a recent study of DNA methylation in DLB by Pihlstrom et al. The issue of the sample size in this study needs to be addressed more adequately. What size effects were you powered to detect? Although Pihlstrom et al. studied a different brain region, it may be useful to perform a look-up of the significant CpGs identified in this study to look for replication in Pihlstrom et al.
3. What was the rationale for analysing beta-values rather than M-values, which have statistically preferable properties (see Du et al., 2010)?
4. The justification for using an FDR correction and not one of the recommended EWAS significance thresholds as the primary approach is unclear. It is not necessary to use an FDR correction in order to provide a resource to researchers as the authors claim, as the same could be achieved using a standard EWAS threshold and providing full summary statistics for all tested CpGs. It is also incorrect to describe the FDR correction as “extremely stringent” or as a compensation for the small sample sizes. I can understand why it might be preferable to perform functional/genomic enrichment analyses using a list of CpG sites determined through a more relaxed significance threshold but a standard EWAS significance threshold (either Saffari et al., 2018 or Mansell et al., 2019) should represent the primary approach for declaring significance. In the discussion, reference is made to the fact that previous studies have applied p-values that do not control for effect size. It is unclear what this means or how this study differs in its approach. Moreover, it isn’t clear (given limited understanding of what differences in DNA methylation mean) whether controlling for effect size would even be desirable.
5. Please include a Q-Q plot for the EWAS in the supplementary materials and comment on it in the results section.
6. The enrichment analysis provides interesting insights into the differentially methylated CpGs. It is possible, however, that enrichment in particular genomic locations or chromatin states is seen simply because these locations are enriched for CpG sites that show variation in a population (and, by definition, the associated CpGs must show variation). That is, since many CpG sites on the array are non-variable, perhaps by using all CpGs on the array as the background set, one simply sees enrichment in areas of the genome/chromatin states that contain variably methylated sites. It might be possible to test this idea by (i) assessing how the sites that are associated with DLB rank in terms of variability compared to the whole array and (ii) selecting a background set that is matched for variability.
7. What happens if the enrichment analysis is performed for hyper- and hypo-methylated sites separately?
8. Did your approach to GO analysis account for biases caused by differences in the number of CpG sites per gene (Maksimovic et al., 2021)? If not, consider using the KEGG and GO tools in missMethyl where it is possible to apply this correction. When reporting the results from the functional enrichment analysis, it needs to be clearer upfront that no associations survived FDR correction (and also how far the best p-value is from being significant after FDR adjustment).
9. “This provides clear evidence that aberrant DNA methylation occurs in subjects with DLB pathology, and some of the methylation sites are likely to be epigenetic mechanisms which contribute to the development of the disease.” It is not possible to assert causality from the results presented in this

manuscript, this claim should be modified or further justified.

10. The approach of using an FDR adjustment to correct for multiple testing cannot be described as “extremely stringent”

Minor comments:

1. In the abstract, state which area of the brain was studied.

2. Make it clear that SNCA is the gene encoding alpha synuclein—you do this later in the introduction but perhaps move the explanation to the first mention of the gene name.

3. “toxin like” should be “toxin-like”

4. “Epigenetic mechanisms including the methylation of CpG dinucleotides on genomic DNA, mediate the gene-environment interactions thought to be involved in the development of the disease”

This sentence is ambiguous: it can either be read as a claim that epigenetic mechanisms mediate the gene-environment interactions that are believed to be involved in DLB (which is quite a strong claim that would definitely need a supporting reference) or that epigenetic mechanisms mediate gene-environment interactions in general, which is, of course, a widely accepted belief. Please make it clear which meaning is correct.

5. “Compared with other neurodegenerative diseases such as PD and AD, where comprehensive studies of DNA methylation regulation have been conducted, very few studies have undertaken this for DLB”

It is true that comprehensive studies of DNA methylation have been carried out for these conditions; however, these studies have generally just identified sites showing differences in methylation rather than showing that the DNA methylation is actively involved in regulating anything. As such, the term “DNA methylation regulation” is misleading. In many cases, the role of DNA methylation is unknown and, in many places in the genome, variation in methylation does not appear to be correlated with gene expression.

6. “Desplats et al. (2011) found global DNA hypo-methylation...”: clarify whether this is the SNCA gene or the whole genome.

7. “ANK1, a well-known epigenetic target of AD”: not accurate to call ANK1 a target of AD as this suggests a direction of effect that is not known.

8. “Gender” should be “sex”

9. “Probes when the last 3 bases in its target sequence overlapped with known single nucleotide polymorphisms (SNPs)”: did you only consider SNPs with minor allele frequency above a certain threshold? Is there a study that supports the use of the three base cut-off? I am aware of the recommendation to use a five base cut-off by Zhou et al. (2017). This sentence also contradicts the first sentence in the “Differential methylation site analysis” section of the results section, where you state “Following the removal of all probes which overlapped with known single nucleotide polymorphisms”. Please clarify the approach used.

10. “overlapped with non-CpG context”: can you clarify what this means?

11. “probes with unreliable measurements (p-value>0.05)”: clarify that this is the detection p-value

12. “along with those probes previously described to hybridize to multiple locations in the genome were removed”. Please cite the reference you used to identify these probes.

13. Table 1: add the unit of measurement for post-mortem interval

14. “These DMCs showed a mild-to-moderate effect size with the average absolute β -difference of 0.07 (sd= 0.021, range from -0.18 to 0.16)”. I am not convinced that this is a useful statement. I don’t think an average beta difference is a particularly useful figure to report, particularly when the causes and consequences of variation in DNA methylation are poorly understood and appear to vary according to

context (i.e. a small difference might be very important in some contexts whilst a large difference might have little effect in others). It's also worth noting that that effect of a given change in beta-value will quite possibly vary depending on where on the scale of possible beta-values this change occurs (i.e. where between 0 and 1).

15. It would be preferable for the figures in Figure 1A to be slightly larger

16. Table 2 should be ordered on p-value as there are ties between the FDR adjusted p-values. Gene names should be italicised (in other tables too).

17. Some words are unnecessarily capitalised (bivalent, zinc)

18. Please clarify the meaning of this: "...the majority were not captured by marginal DLB associations and may represent important, additional loci for DLB related epigenetic changes"

19. Use "methylome-wide association study" or "epigenome-wide association study" rather than "genome-wide association study"

20. The meaning of "intelligence related diseases" is not clear

21. "Our study corroborates some previously reported genes that were associated with DLB". Explain how these genes have been implicated in DBP (i.e. GWAS or EWAS etc)

22. The findings re. PAK6 are interesting: is there any evidence linking altered methylation at the associated sites to an effect on expression of PAK6?

23. "To the best of our knowledge, this is the first reported DNA methylation association analysis to have been conducted on DLB brain tissue". How does this relate to the study of Pihlstrøm et al., which is also discussed?

24. The bias towards finding hypomethylation in the DLB cases is an interesting finding. How does this relate to the findings of Pihlstrøm et al.? Are there any studies of gene expression in DLB (preferably brain) that also show a bias in the direction of the associated gene expression changes?

Response to Reviewer Comments

We thank the three Reviewers for the helpful and constructive comments to help us improve the study. Please find below our point-by-point response to the comments raised by each of the Reviewers. We have reanalyzed the data and revised the manuscript accordingly. Questions and comments of the Reviewers are numbered below, our response are in blue. We marked changes to the text using Track Changes in the updated version of our manuscript.

Reviewers' comments:

Reviewer #1 (Remarks to the Author):

The work looks at the DNA methylation profile of 31 post mortem brain samples from Dementia with Lewy bodies (DLB) patients and controls using the Illumina EPIC array. Overall this is a well written manuscript with some interesting points in a new area as very little work on the epigenetics of DLB has been published to date. However, I do have some fundamental issues with the results that means at this stage I cannot recommend publication.

- Does the manuscript have technical or conceptual flaws that should prohibit its publication? If so, please provide details.

This study has a small sample size for an epigenome wide association study. Despite this, the authors report detection of 296 differentially methylated sites and 215 differentially methylated regions (DMRs), this is far in excess of what would be expected for a sample cohort of this size. Potential reasons for this could be:

1. A lack of sample randomisation in the running of the EPIC arrays, the manuscript does not state randomisation occurred. This would cause huge batch effects that would not be correctable through the use of surrogate variables (SVs).

Response: Thanks for your comments. We acknowledged that there were batch effects in our study. Indeed, due to the way samples were obtained, the DLB cases and controls were analyzed on different batches. When we analyzed the genome-wide data, there was indeed inflation in the statistics consistent with batch effects. We thank the reviewer for highlighting the question, since in fact on further investigation it was evident that SVA did not capture these batch effects optimally and in fact was probably partially capturing the associations of interest. Following the reviewer's comments, we investigated a number of possible ways of minimizing the inherent confounding due to the study design, and contacted a statistical genetics expert (who has been added to the list of authors). Therefore, we replaced SVA by PCs built on the negative control probes, to try and capture batch effects without also capturing signal. The top three PCs were selected based on a scree plot, see Figure below (also as the **Supplementary Figure 1**) for the scree plot of the PCA analysis, and the first two PCs showed significant associations with EPIC chip slides by the Kruskal-Wallis rank sum test (p -value = 0.004 and 0.0003, respectively). Based on the genomic inflation calculation, these negative control probe PCs corrected models show a smaller inflation of p -values from expectation. For all of the following responses, we have applied this method to all of our results.

Negative controls probes

Supplementary Figure 1. Scree plot of the negative control probe-based PCA analysis.

2. Inflation lambda statistics should be included to show that the findings of this study are not over inflated.

Response: See also our response to the previous point. We now have included the qq-plot with lambda statistics as indicated with the new approach to correcting for batch effects. The updated model was adjusted by sex, age, PMI, Neuronal cell proportions and three top PCs from the negative control probes, and the corresponding genomic inflation (lambda) is 1.1, indicating that the model was not overly inflated. The qq-plot and Manhattan-plot are included in **Figure 1** of the revised manuscript.

Figure1. The distribution of differentially methylated CpGs. Left panel: QQ-plot of the p-values from the EWAS model. The genomic control inflation rate of the EWAS model is 1.1, indicating a modest inflation. The x-axis shows the expected $-\log_{10}$ (p-value), whereas the y-axis indicates the observed $-\log_{10}$ (p-value). **Right panel:** Manhattan plot of the EWAS results between DLB and Control samples. The red line indicates the genome-wide significance threshold (p-value = $9e-8$) and blue line represents a q-value of 5 % adjusting for multiple comparisons. The top DMCs per chromosome are annotated.

3. Detail on the criteria of DLB pathology for sample inclusion is missing from the manuscript. From post-mortem tissue only it is not possible to determine DLB from Parkinson's disease. How are the authors confident that this doesn't bring variation into their results?

Response: Thank you for your comment. You are correct, it is not possible. However, all samples were diagnosed based on archival medical history in conjunction with neuropathology reports.

We have added the following sentences in the method section (line 122-125):

"We evaluated DLB post-mortem brain based all medical history on archival information (albeit limited) and neuropathology reports. Control brains were selected from subjects who had no clinical history of dementia, few or no neuritic plaques, a Braak tangle stage of III or less and no other significant neuropathological abnormalities.

4. Similarly the criteria for control sample inclusion is equally vague. The authors state the use of "non-cognitively impaired controls". How was cognitive ability assessed and were any pathology levels measured in control tissue?

Response: The control brains were selected from subjects who had no clinical history of dementia, few or no neuritic plaques, a Braak tangle stage of III or less and no other significant neuropathological abnormalities. We have added this information in the method section.

5. Brain region studied seems inconsistent, originally stated as Neocortex, Brodmann area 9, but later stated as only Parietal Cortex. As the Neocortex is a large area of the human brain caution is needed in interpreting results as this would introduce brain region variation in the methylation data. Could the authors please clarify that all samples were from the same brain region?

Response: We have confirmed with the Brains for Dementia (Brain Bank, UK) that all the samples were from the same brain region, that is, the Neocortex and Brodmann area 7. We have removed "parietal cortex" in the discussion section to make it clear.

In addition fundamental QC metrics are missing from the methods including:

6. Signal intensity check of both methylated and unmethylated channels.

Response: We now have provided the missing information in the Methods section. We have added the signal intensity plot as **Supplementary Figure 2A**.

Supplementary Figure 2. Quality control of DLB EPIC array data. (A). Signal intensity plot of all the samples. Median intensity of methylated and unmethylated channels intensity per sample was calculated. (B). Sex prediction based on the median total intensity on sex-chromosomes. (C) Hierarchical clustering of samples with SNP profiles.

7. Sex check and hierarchical clustering methods (based on RS probes) to be sure of no sample mix ups

at the array stage.

Response: We thank for the reviewer's comment! We note one sample was mislabelled and we corrected it. Now we have reported no sex mix-ups for this dataset (**Supplementary Figure 2B**). We do not have genotype data for this dataset, so cannot perform sample mix-up comparisons by comparing to the SNP probes on the EPIC array. Nonetheless, we now have provided the hierarchical clustering plot based on the SNPs identified on the EPIC array as **Supplementary Figure 2C**. We did not observe any outliers based on the genetic similarity.

8. There was also no note in the methods listing how many samples/ probes were lost due to p filtering, cross-hybridising, SNP probe removal etc.

Response: No samples were removed at the QC stage. Previously we used RnBeads to perform the analysis, where some details were a bit difficult to extract due to the feature of wrapped functions in the pipeline. Instead of detangling all the detailed codes in the RnBead software, we decide to switch software and have now used the minfi and missMethyl R packages to redo the analysis. We are now able to add details on numbers of filtered probes at each step of the QC in the results section. We have added the following paragraph in the result section (line 238-253):

“We first performed a series quality control filters on the Illumina's EPIC probes (see the Method for the details). We started with 865,859 CpG probes that are mappable to the human genome (version hg19) based on the EPIC annotation file (R package: IlluminaHumanMethylationEPICanno.ilm10b4.hg19, version 10.b4), 10,841 poor quality probes (i.e., detection p-value is not less than 0.01 in all samples) were removed. 28,260 probes were further removed as they were affected by SNPs (either at the CpG interrogation site or at the single nucleotide extension). In addition, 40,565 and 2,576 probes were also excluded that demonstrated cross-reactivities and overlapped with non-CpG context (i.e. CpA, CpC and CpT). Finally, 17,209 probes located on the sex chromosomes were deleted. Subsequently, 766,399 CpGs were used for downstream analysis after removing an additional 9 CpGs which contained missing data. Of note, no samples were removed due to quality concerns (**Supplementary Figure 2A**). Furthermore, the sex chromosome probes were used to estimate the sample's sex, which matched reported sex for all samples (**Supplementary Figure 2B**). Furthermore, hierarchical clustering was performed using the SNP probes to demonstrate genetic similarities among samples and no outliers were observed based on the genetic similarities (**Supplementary Figure 2C**).”

9. Following the methylation data analysis the authors make use of ChromHMM data from Roadmap Epigenomics. This data was from prefrontal cortex only and not neocortex or parietal cortex as used in this study. Therefore any conclusions drawn from the interaction of DNA methylation and chromatin state is likely overstated.

Response: Indeed, we agree wholeheartedly with the reviewer. As such, we have acknowledged that the ChromHMM data from Roadmap was from prefrontal cortex and not the same tissue as used for EPIC array (i.e., Neocortex), but this is the closest one we could obtain. We have mentioned this limitation in the method section (line 197-203):

“We noted that we have used the chromatin states inferred from the prefrontal cortex of the brain rather than the neocortex, since chromatin state inferences for the latter are not currently available

from the RoadMap Epigenomic Project. Thus, the enrichment results should be treated with a certain level of caution and considered suggestions rather than accurate predictions. However, given the physical proximity of the prefrontal cortex to the neocortex, we believe the enrichment analysis provides results of biological relevance regarding the regulatory regions.”

10. There were few grammatical and spelling errors in the manuscript and all tables, but particularly table 2 would benefit from a clear legend.

Response: Sorry for the errors. We have carefully checked them and revised accordingly. We now have added detailed legends to all the tables.

11. • Are the conclusions original? If not, please provide relevant references.

I find the conclusions to be original and interesting take on the field however given the highly questionable significance of the data presented the conclusions are likely over stated.

Response: We have now reanalysed the datasets and drawn our conclusions carefully. We have added relevant references to support what we have observed and highlight the new discoveries.

12. • Do you feel that the results presented are of immediate relevance for people in your own discipline or for a broader audience? If you recommend publication, please outline briefly what you consider to be the outstanding features.

The results of this study would be of interest to the DLB field generally however in its current state I cannot recommend publication.

Response: We hope that the reviewer will reach a different conclusion after reading the revised version.

13. • If you feel that specific additional experiments would strengthen the case for publication in Communications Biology, please provide suggestions.

I would strongly recommend a re-analysis of this data to determine if the findings are real and not a result of batch effects or other factors in the experimental design. It would also be of benefit to replicate these findings in a targeted manner using an alternative technology, for example bisulfite pyrosequencing.

Response: As recommended, we have performed a complete re-analysis of the study to minimize the potential influence of batches or other experimental artifacts by including several principal components built from the signals among the negative control probes. We followed up our analysis by a literature review of identified genes and functional enrichment analyses. We acknowledge the importance of performing a validation study in a new dataset, but given how rare these samples are, and how challenging it is to obtain such samples from DLB patients, we must leave independent validation for future work. Meanwhile, our previous study and those completed by others have indicated that in general the methylation signals detected either by Illumina array or WGBS sequencing were well replicated by the bisulfite pyrosequencing.

Reviewer #2 (Remarks to the Author):

The authors of this study are examining methylation indices in cases of dementia with Lewy bodies. This

is an unmet need in the field and influenced from underwhelming progress of genome wide association studies. The study results are an important step forward towards understanding the pathogenesis of dementia that in actuality - represent an oversimplified understanding of the influence of genomic methylation and its relevance towards biomarker validation.

Main points.

1. Abstract: this is poorly described with the reader having to guess what type of technology used to distinguish DNA methylation. Furthermore, the number of DLB cases (15) and NCI controls (16) are not described. Likewise, adding effect sizes and statistical significance to findings would be helpful.

Response: Thank you for your suggestions. We have now revised the abstract by adding the requested information. However, due to the word limitation (i.e., 150 words), we cannot add much information regarding the effect size and statistical significance etc.

2. Statistical considerations: Study size is without question small and difficult to determine how broad the methylation signatures identified would replicate in larger DLB populations. Is the study too underpowered and what are the power calculations?

Response: We acknowledge the limitation of our small sample size and note this in the manuscript. However, we would like to note that this is a pilot study with rare, hard-to-obtain, high-quality DLB brain samples, and we did observe some interesting results which we think will provide novel and insightful knowledge to the field.

Regarding the power calculation, we have applied the R package 'pwrEWAS' which was designed to estimate the power specific to EWAS studies (Graw et al. BMC Bioinformatics, 2019). As there is no brain tissue reference methylome available in the 'pwrEWAS' package, we thus explored multiple tissues included in the package with hope to have an approximate estimation for brain. To mimic the scenario we have in this study, we selected balanced sample size per group (i.e. sample rate for group 1 is 0.5), the number of CpGs to be tested as 800,000, multiple testing correction FDR threshold as 0.05 and target number of DMCs as 30. Since we do not know the likely range of effect sizes for the DLB associated DMCs, we have calculated power using four values for the mean methylation difference between two groups. Specifically, we used 0.1 (which is close to the average effect size of our detected DMCs), 0.2, 0.3 (close to the maximum effect size among our detected DMCs at q -value < 0.05) (See **Supplementary Figure 3A** for the effect size distribution of our DMCs at q -value < 0.05) and 0.4 (one of the effect sizes reported in Pihlstrøm et al.). With 31 samples, we achieve power ranged from 0.6 to 0.8 for these four effect size setting for most of the tissues. Overall, our power calculations suggest that we would need 40-50 samples to achieve 80% power for the smaller effect sizes (**Supplementary Figure 3B-F**). In this way, our sample size in this study is slightly underpowered.

Supplementary Figure 3. Effect size and power calculation. (A). Effect size ($\Delta\beta$) distribution of DLB-associated DMCs at q-value < 0.05. Power estimation against sample size with different effect sizes using pwrEWAS with different tissues as reference tissue methylome: (B) Adult PBMC; (C) Adult blood; (D) Sperm; (E) Saliva and (F) Liver.

We have now added the following sentences to the method section (line 178-186):

“Lastly, we adopted the R package “pwrEWAS”⁴¹, which was designed to estimate the power specific to EWAS studies, to perform the power analysis. It provided the power estimation for the most commonly used tissue types for EWAS but no brain tissue. We thus explored multiple tissues included in the package with hope to have an approximate estimation for brain. To mimic the scenario we have in this study, we selected balanced sample size per group (i.e., sample rate for group 1 is 0.5), the number of CpGs to be tested as 800,000, multiple testing correction FDR threshold as 0.05 and target number of DMCs as 30. Since we do not know the likely range of effect sizes for the DLB associated DMCs, we have calculated power using four values for the mean methylation difference between two groups: 0.1, 0.2, 0.3 and 0.4.”

And the sentences to the discussion section:

“Firstly, we do recognize the small sample size used to conduct our study. Specifically, power calculations suggest that we would need 40-50 samples to achieve 80% power for effect sizes ranged between 0.1 and 0.4 which covered the mean effect size and maximal effect size detected in our study (**Supplementary Figure 3B-F**). In this way, our sample size in this study is slightly underpowered.”

Graw, S., Henn, R., Thompson, J. A. & Koestler, D. C. PwrEWAS: A user-friendly tool for comprehensive power estimation for epigenome wide association studies (EWAS). *BMC Bioinformatics* **20**, 1–11 (2019).

3. Have the authors accounted for methylation differences that are known to be associated with age and sex? What other clinical variables exist for the cohort, such as smoking, medication, etc? These details if available should be included in the study for readers to assess.

Response: We have corrected our association analysis with covariates for sex and age. We do not have smoking, info on polypharmacy etc. We asked the brain biobank in the UK that provided the samples whether they had such information and they do not. We have checked for overlap between our DMC list and the reported smoking-CpGs as well as age-CpGs (i.e., Horvath’s 353 epicloek CpGs), and did not see any overlap.

4. What is the significance of reduced DLB methylation? Do the authors believe hypomethylation converge on transcription factor binding sites (TFBS) to regulate gene expression changes? Are TFBS represented in the WGCNA? Are the authors proposing reduced methylation with DLB pathogenesis?

Response: Thank you for this interesting comment. We have now re-analyzed the datasets and reported the DMCs with a q-value < 0.05. With this setting, we observed 60% of the DMCs were hypo-methylated. We hypothesize that these hypo-methylation events at multiple genes might be related to the notable reductions in nuclear levels of Dnmt1. However, to test this hypothesis we would need to perform further functional validation studies.

Since we do not have corresponding gene expression data, it is not possible to see whether hypo-methylation might regulate gene expression by affecting the transcription factor regulation. However, indirect enrichment analysis in transcription factor targets on different modules detected by WGCNA indicated that the one of the WGCNA modules (the greenyellow one) associated genes were regulated by TF of *NF1*, *ZNF140* and *DNMT3A* etc.

In a blood DNA methylation paper (Nasamran et al. *Alzheimer's Dement.*, 2021), the authors found reduced DLB methylation compared with Parkinson's disease dementia (PDD). Also, hypo-methylation was observed in multiple genes including *SNCA*, *APP* etc. in other DLB studies. These observations indicate that reduced methylation at certain genes might be associated with DLB pathogenesis or progression. However, these hypotheses would need further functional investigation.

Nasamran, C. A. et al. Differential blood DNA methylation across Lewy body dementias. *Alzheimer's & Dementia: Diagnosis, Assessment & Disease Monitoring* 13, e12156 (2021).

5. The authors should specifically discuss the study limitations clearly.

Response: We have acknowledged the limitation of our small sample size and discussed the potential influences of our conclusion due to that, as well as other limitations.

We have added the following paragraphs to the discussion section (line 496-531):

"In addition to the observations reported herein and the obvious strengths of our study, we do note this study's limitations. Firstly, we do recognize the small sample size used to conduct our study. Specifically, power calculations suggest that we would need 40-50 samples to achieve 80% power for effect sizes ranged between 0.1 and 0.4 which covered the mean effect size and maximal effect size detected in our study (**Supplementary Figure 3B-F**). In this way, our sample size in this study is slightly underpowered. Nevertheless, these samples are rare and difficult to obtain: consenting, harvesting, and storing the samples are all very challenging endeavours for DLB neocortex. Although we didn't replicate most of the previously reported DLB signals at stringent significance thresholds (q -value < 0.05 or p -value $< 9e-8$), we were able to detect several new signals/genes reported to be associated with other neurodegenerative diseases such as PD and AD. Another limitation of our study is the inability of the standard sodium bisulfite conversion method used in the MethylationEPIC array analysis to distinguish between 5-methylcytosine (5mC) and 5-hydroxymethylcytosine (5hmC), with the latter being enriched in brain cells and thought to play an important role in neural development and neurodegenerative diseases^{72,73}. Parallel profiling of methylation and hydroxymethylation⁷⁴ or oxidative bisulfite sequencing⁷⁵ might be feasible solutions for capturing intermediary DNA methylation states between methylated and unmethylated cytosine.

We acknowledge the limitation due to the design, whereby the DLB and control samples were run on different batches of arrays from Illumina. We used principal components (PCs) constructed from the negative control probes to minimize confounding by batch. Approaches such as surrogate variable analysis, which adjust for the case-control status while calculating the surrogate variables from the majority of probes, led to QQ plots showing substantial inflation of significance. Although we realize that any attempt at correction may eliminate some true signals since the DLB samples and control samples were run on different chips and batches, correcting the model using these PCs (based only on the negative control probes) resulted in better distributional results than other corrections that we explored. We expect this strategy to be fairly conservative since the negative control probes have low

fluorescent signals and therefore, this correction may not capture batch effects in high signal ranges. It is worth noting that we previously normalized the methylated and unmethylated signals with funNorm, which uses all types of control probes to align the signals across arrays. After detailed consideration of various options, we decided to implement this relatively conservative correction rather than use a method that will eliminate potential signals of interest. Although we performed functional enrichment analyses on a large set of DMCs with a relaxed significance criteria (i.e. q-value < 0.25, instead of using q-value < 0.01 or < 0.05), the analyses showed certain interesting results which are relevant to DLB and other neurodegenerative diseases. Overall, a larger sample size and meta-analysis with other cohorts would help corroborate the findings reported herein.”

6. Cell type considerations: the influence of cell-type specific DNA methylation is briefly described. The authors should be considering analytical methods like Cell DMC or equivalents and discussing results with/without cell type adjustment. What is the impact of different cell type influences such as estimateCellCounts2 or methylCIBERSORT?

Response: We have compared the analysis results without cell-type adjustment and observed that without cell-type proportion correction, we observed larger inflation of significance (i.e., $\lambda=1.21$ vs 1.1 with cell-type proportion correction). Meanwhile, the correlation between the two sets of p-values is only moderate ($\text{cor}=0.61$), see Figure below. We note that the top signals are quite consistent between two models.

Figure. Correlation between two models with and without NeuN proportion correction.

We have added the following sentences to the Discussion section (line 474-478):

“Meanwhile, we have also compared the models with and without estimated neuronal cell proportions as co-variables. We observed that the inflation lambda is much larger for the model without estimated neuronal cell proportion correction (i.e. lambda=1.21) and the correlation of p-values between the two models is moderate (cor =0.61) (**Supplementary Figure 10B**).”

We were not able to run the estimateCellCounts2 function on the EPIC array for the DLPFC tissue. There was no required package “FlowSorted.DLPFC.EPIC”. It seems there were no DLPFC tissue reference probes in the EPIC array, and it will eventually reduce to 450K array data which will be the same as estimateCellCounts function. We tried methylCIBERSORT, used the population methylation data from the R package “FlowSorted.DLPFC.450K” and selected 100 cell-type specific CpGs through the default setting of the feature selection function in methylCIBERSORT package. We then used methylation profiles of these 100 cell-type specific CpGs as the signature matrix to estimate the neuronal proportion through the CIBERSORTx website. When applying the CIBERSORTx estimated neuronal proportion as a covariate to correct the DLB association model, we obtained the genomic inflation (lambda) of 1.26, indicating a larger inflation than the model corrected by the cell proportion estimated by estimateCellCounts function. Meanwhile, the correlation between $-\log_{10}(\text{p-values})$ of the two models

(M1: estimateCellCounts-based; and M3: methylCIBERSORT-based) is moderate ($r=0.57$). Although the models show overall poor agreement, the rank of top signals (e.g. those with p -value $< 1e-5$) seem consistent.

Figure. Correlation between two models with different estimated NeuN proportion correction.

7. While the methylation data is interesting the above questions should be considered in light of dementia pathogenesis and is relevant to biomarker validation.

Response: We now have addressed all the questions above.

Reviewer #3 (Remarks to the Author):

1. This study represents a thorough assessment of differences in DNA methylation Brodmann area 7 of the post-mortem brains of patients with Dementia with Lewy Bodies (DPB) and controls. Several differentially methylated sites and regions are identified, with several of these sites and regions identifying genes that are interesting and plausible candidates for DLB. The authors make the interesting observation that there is a bias towards identifying hypomethylation in the cases and they link this to a previously observed reduction in nuclear Dnmt1 in the post-mortem brains of patients with DLB and a mouse model of the condition. Further follow-up analyses to explore this bias would be of interest and

could be performed by tweaking the existing functional and genomic enrichment analyses. Weighted gene coexpression network is performed and represents a useful and informative extension to the other enrichment analyses that are performed.

Response: Thanks for summarizing our study.

2. These findings are interesting and novel, being one of the first post-mortem explorations of DLB. The main limitation of the study is the relatively small sample size. I do not feel this is adequately addressed either in terms of presenting power calculations to attempt to identify the magnitude of the methylation differences that are possible to identify in this sample or simply acknowledging the effects of the small sample in the discussion. Since this is an early study of post-mortem DNA methylation in DLB and the first to study this particular brain region, I think the results are likely to be of broad interest but the limitations of the sample size must be fully addressed.

Response: Thanks for your comments. We acknowledged the limitation of our small sample size. We have added it in the discussion. Please refer to the answers to the question 2 of Reviewer 2. Particularly, we have added the following sentences to the Discussion section (line496-503):

“In addition to the observations reported herein and the obvious strengths of our study, we do note this study’s limitations. Firstly, we do recognize the small sample size used to conduct our study. Specifically, power calculations suggest that we would need 40-50 samples to achieve 80% power for effect sizes ranged between 0.1 and 0.4 which covered the mean effect size and maximal effect size detected in our study (**Supplementary Figure 3B-F**). In this way, our sample size in this study is slightly underpowered. Nevertheless, these samples are rare and difficult to obtain: consenting, harvesting, and storing the samples are all very challenging endeavours for DLB neocortex.”

3. The manuscript is fairly clearly written, although some grammatical errors (e.g. missing commas) hamper its readability in places. Some additional details are required in the methods section to permit reproduction of the analyses (I note these below). In general, the statistical analyses appear sound, although I have some comments below about the unit of methylation measurement used for analyses and the choice of significance threshold and claims made regarding the FDR correction and the small sample size. The conclusion would benefit from some re-writing to make sure the main messages and limitations of the study are clearly and concisely presented.

Response: We have re-analysed the dataset and revised the manuscript carefully following all the reviewer’s comments. Details are provided below.

I have made some specific comments below:

Major comments:

1. “Such mechanisms regulate the transcriptomes of neuronal cells, and play important roles in neurodegeneration.”. Please provide citations to support these claims. It would also be good to provide more specific information about DNA methylation, since this is the focus of the study.

Response: Thanks for your comments. We now have added relevant references:

Kwok, J. B. Role of epigenetics in Alzheimer’s and Parkinson’s disease.

<http://dx.doi.org/10.2217/epi.10.43> 2, 671–682 (2010).

Feng, Y., Jankovic, J. & Wu, Y. C. Epigenetic mechanisms in Parkinson’s disease. *Journal of the*

Neurological Sciences 349, 3–9 (2015).

Landgrave-Gómez, J., Mercado-Gómez, O. & Guevara-Guzmán, R. Epigenetic mechanisms in neurological and neurodegenerative diseases. *Frontiers in Cellular Neuroscience* 9, 58 (2015).

We have also added additional specific information about DNA methylation in neurodegenerative diseases (line 72-76):

“For instance, Humphries et al. integrated whole transcriptome and DNA methylation analysis identified disruptions in both DNA methylation and transcription with genes in the myelination network which related to synaptic function and behavioral response in both late-onset Alzheimer’s disease (LOAD) and DLB²².”

Humphries, C. E. et al. Integrated Whole Transcriptome and DNA Methylation Analysis Identifies Gene Networks Specific to Late-Onset Alzheimer’s Disease. *Journal of Alzheimer’s Disease* 44, 977–987 (2015).

2. “These studies are extremely limited because they are either based on targeted candidate genes or were statistically underpowered”. Please can you elaborate on why you think they were underpowered? Can you also set out in the description (in the next paragraph) of your study why it is adequately powered/better powered than previous efforts? The sample sizes studied in this study are small for an EWAS and small compared to those studied by a recent study of DNA methylation in DLB by Pihlstrom et al. The issue of the sample size in this study needs to be addressed more adequately. What size effects were you powered to detect? Although Pihlstrom et al. studied a different brain region, it may be useful to perform a look-up of the significant CpGs identified in this study to look for replication in Pihlstrom et al.

Response: The limitation mentioned in this sentence was specifically aimed at previous studies either focusing on candidate genes (e.g. Desplats et al. 2011, Tsuchida et al. 2018, and Ozaki et al. 2020) using pyrosequencing or using WGBS but only profiling on two samples (e.g. Sanchez-Mut et al. 2016). The “underpowered” thus was specific to the study of Sanchez-Mut et al. 2016. We have change the “were statistically underpowered” to “or were only analyzed on couple of samples”. During the preparation of this manuscript, we were aware of Pihlstrom et al’s study and had compared our DLB DMC list with their results. Unfortunately, we did not observe overlaps, possibly due to the study of different brain regions as well as different comparisons. To clarify, we have now added an additional sentence to the beginning of the subsequent paragraph by introducing Pihlstrom et al’s work and removing “for the first time” (line95-100):

“Very recently and during the preparation of our study, Pihlstrom et al. performed a large epigenome-wide association study (EWAS) in a cohort of DLB post-mortem human brain tissue (frontal-cortex; n=332) to identify DNA methylation changes associated with the disease. Like Pihlstrom et al., this study aims to demonstrate the power of profiling genome-wide DNA methylation sites in pathologically confirmed cases of DLB but focusing on a different brain region. “

We acknowledged the relative small sample size we collected for this DLB epigenetic study, and added relevant discussions. Regarding the power calculation, please also refer to our response to Reviewer two’s question 2.

3. What was the rationale for analysing beta-values rather than M-values, which have statistically preferable properties (see Du et al., 2010)?

Response: We totally agree with the reviewer. Actually, we applied the M-values for the differential methylation analysis and used beta-values for the visualization and methylation difference calculation as the beta-value is more intuitive to understanding the DNA methylation effects.

Although Du et al. (2010) suggested that the M-values based model would have statistically preferable properties, there is no general consensus whether to use beta or M values for statistical analyses of DNA methylation array data as discussed in the Pihlstrøm et al. (2021). Actually, Pihlstrøm et al. (2021) pointed out that the beta-value based method has been favored in previously published studies on AD neuropathology.

Nevertheless, we have tried both models and there was no significant differences. We mentioned the comparisons in the Discussion section (see below the new sentences added to the Discussion section, line 468-474). We have also provided the full CpGs list with p-value < 1e-3 for both models in **Supplementary Table 4 & 5**.

“We also conducted the EWAS analysis based on beta-values. The genomic control inflation lambda for the beta model is 1.06, which is slightly better than the M-value model. Overall, the signals detected by the two models are very similar and the p-values are highly correlated (Pearson correlation cor=0.96) (**Supplementary Figure 10A**). In the interests of providing the research community with a more comprehensive resource, we have provided a list of DMCs with a p-value < 1e-3 for both models (**Supplementary Table 4 & Table 5**).”

4. The justification for using an FDR correction and not one of the recommended EWAS significance thresholds as the primary approach is unclear. It is not necessary to use an FDR correction in order to provide a resource to researchers as the authors claim, as the same could be achieved using a standard EWAS threshold and providing full summary statistics for all tested CpGs. It is also incorrect to describe the FDR correction as “extremely stringent” or as a compensation for the small sample sizes. I can understand why it might be preferable to perform functional/genomic enrichment analyses using a list of CpG sites determined through a more relaxed significance threshold but a standard EWAS significance threshold (either Saffari et al., 2018 or Mansell et al., 2019) should represent the primary approach for declaring significance. In the discussion, reference is made to the fact that previous studies have applied p-values that do not control for effect size. It is unclear what this means or how this study differs in its approach. Moreover, it isn't clear (given limited understanding of what differences in DNA methylation mean) whether controlling for effect size would even be desirable.

Response: We note that we reported the DMCs following one of the recommended EWAS significance thresholds (Mansell et al., 2019) in our previous version. In the updated version with the re-analyzed dataset, we report two significant DMCs at a threshold of p-value < 9e-8 (adjusted p-value < 0.02).

We completely agree with the reviewer that the FDR correction itself doesn't mean “extremely stringent”. In the previous version of the manuscript, we highlighted “extremely stringent” when requiring minimum methylation differences $\geq 5\%$ compared with only using FDR criteria. One of the considerations of requiring minimum methylation differences was to ensure the model with limited sample size in our study would have good power for detection. Nevertheless, we have revised it in the revised script and removed the effect size controlling.

We have removed the sentence which highlighted the effect size controlling of our study compared with other studies. In the new version, we applied the q-value or recommended p-value only.

5. Please include a Q-Q plot for the EWAS in the supplementary materials and comment on it in the results section.

Response: Following the reviewer's suggestion, we have added the Q-Q plot in **Figure 1** and mentioned it in the result section (line257-258): "We note that this model returned a genomic control inflation (GCin) of 1.1 (**Figure 1A**), indicating a modest inflation."

Figure1. The distribution of differentially methylated CpGs. Left panel: QQ-plot of the p-values from the EWAS model. The genomic control inflation rate of the EWAS model is 1.1, indicating a modest inflation. The x-axis shows the expected $-\log_{10}(p)$, whereas the y-axis indicates the observed $-\log_{10}(p)$. **Right panel:** Manhattan plot of the EWAS results between DLB and Control samples. The red line indicates the genome-wide significance threshold (p -value = $9e-8$) and blue line represents a q-value of 5 % adjusting for multiple comparisons. The top DMCs per chromosome are annotated.

6. The enrichment analysis provides interesting insights into the differentially methylated CpGs. It is possible, however, that enrichment in particular genomic locations or chromatin states is seen simply because these locations are enriched for CpG sites that show variation in a population (and, by definition, the associated CpGs must show variation). That is, since many CpG sites on the array are non-variable, perhaps by using all CpGs on the array as the background set, one simply sees enrichment in areas of the genome/chromatin states that contain variably methylated sites. It might be possible to test this idea by (i) assessing how the sites that are associated with DLB rank in terms of variability compared to the whole array and (ii) selecting a background set that is matched for variability.

Response: Thank you for your comments. We acknowledge that the DLB associated CpGs may fall among the more variable methylation sites. Therefore we did some additional analyses. See Figure below (**Supplementary Figure 5**) that shows that DLB-associated DMCs showed quite substantial variability, but were not necessarily the most variable ones.

Supplementary Figure 5. The standard deviation (SD) distribution of DLB associated DMCs and top variable CpGs.

The DLB associated DMCs were roughly among the top 20% variable CpGs and the range of sd of the DLB associated DMCs are larger than those top variable CpGs.

We have added the following sentences to the Results section (line 324-326):

“We noted that these DMCs showed certain levels of population variation but not necessarily among the most variable ones (**Supplementary Figure 5**).”

Following your suggestion, we have also tried the genomic feature enrichment analysis of DLB associated DMCs (at q-value <0.25) with different background sets, see the figure below. We do observe that the fold-change were showing decreasing trends for most of the genomic regions when using CpG background sets with increased variations except the CpG shelves and 3’UTR (**Figure A**) but most of the significant ones remain significant even with most top variable CpGs as background, particularly for the enrichment of the first exons. Please note if we use the background set that is matched for variability (i.e., top 20%), the enrichment patterns are closer to the one that used all CpGs as the background. We observed decrease or increase trends of fold-change for most of the Chromatin states when using CpG background sets with increased variations depends on which states it overlapped (**Figure B**). But again, enriched terms were quite consistent for the background set of all CpGs or top 20% variable CpGs.

Figure. Genomic features enrichment analysis for DMCs with different background sets that have different variabilities. (A). Genomic feature enrichment analysis. (B). Chromatin state enrichment analysis.

7. What happens if the enrichment analysis is performed for hyper- and hypo-methylated sites separately?

Response: Thanks for your suggestion. We detected few DMCs, and therefore splitting the DMCs into hypo- and hyper-methylated sites might further reduce the robustness of the enrichment analysis. Nevertheless, we have performed the genomic feature enrichment analysis for hyper- and hypo-methylated sites at q-value < 0.25, separately. Similar to the combined DMCs enrichment result, both hyper- and hypo-DMCs are enriched in the CGI and promoter regions as well as the first exons. Meanwhile, there are differences of the enriched terms for hyper- and hypo-methylated CpGs for different genomic regions, particularly for the TSS flanking regions, bivalent/poised TSS regions and their flanking regions (see Figure below). That is, the enrichment is enhanced for the hypo-methylated DMCs.

Figure. Genomic features enrichment analysis for both hyper- and hypo-methylated CpGs. (A). Genomic feature enrichment analysis. **(B).** Chromatin state enrichment analysis.

8. Did your approach to GO analysis account for biases caused by differences in the number of CpG sites per gene (Maksimovic et al., 2021)? If not, consider using the KEGG and GO tools in missMethyl where it is possible to apply this correction. When reporting the results from the functional enrichment analysis, it needs to be clearer upfront that no associations survived FDR correction (and also how far the best p-value is from being significant after FDR adjustment).

Response: Thanks for your suggestion. We acknowledged this potential issue in our current enrichment analysis where the associated genes were used as the input and we did not consider the number of CpG sites. We have tried the missMethyl and added the related enrichment analysis result to the Results section, see below (line 360-377).

“To account for the potential biases caused by the differences in the number of CpGs per gene, we performed a gene set enrichment analysis on the DMCs and DMRs associated genes using GOMeth and GOREGION functions in the R package missMethyl⁴⁴. Although we used a small numbers of DMCs with a q-value < 0.05, the enrichment analysis returned several neuron development relevant GO terms such as ‘corticospinal neuron axon guidance through spinal cord’, ‘corticospinal neuron axon guidance’, and ‘cerebral cortex tangential migration using cell-cell interactions’ as well as KEGG pathways of ‘neuroactive ligand-receptor interaction’, ‘metabolic pathways’, and ‘pathways of neurodegeneration – multiple diseases’ (**Supplementary Figure 6A**). Meanwhile, when we performed the enrichment analysis on genes of DMCs with a q-value<0.25, the ‘protein maturation by iron-sulfur cluster transfer’ was also

observed as the top enriched GO term. Other top enriched GO terms include 'T cell homeostasis' and 'neurotrophin signaling pathway'. Interestingly, these genes were also enriched in the KEGG pathways of 'ubiquitin mediated proteolysis', 'Ras signaling pathway' and 'prolactin signaling pathway' as well as 'cytosolic DNA-sensing pathway' (**Supplementary Figure 6B**). Furthermore, the analysis revealed that DMR associated genes were enriched in 'regulation of parathyroid hormone secretion', 'regulation of synaptic activity' and highly overlapping with genes involved in 'phospholipase D signaling pathway', 'adrenergic signaling in cardiomyocytes' and 'Parkinson disease' (**Supplementary Figure 6C**).

9. "This provides clear evidence that aberrant DNA methylation occurs in subjects with DLB pathology, and some of the methylation sites are likely to be epigenetic mechanisms which contribute to the development of the disease." It is not possible to assert causality from the results presented in this manuscript, this claim should be modified or further justified.

Response: We agree that the association analysis cannot reveal the causality. We have rewritten this sentence (line 427-430):

"This provides independent supportive evidence that aberrant DNA methylation occurs in subjects with DLB pathology, and some of the methylation sites are likely to be epigenetic mechanisms which may contribute to the development of the disease."

10. The approach of using an FDR adjustment to correct for multiple testing cannot be described as "extremely stringent"

Response: Thanks. As responded in Q4, we highlighted "extremely stringent" when requiring minimum methylation differences $\geq 5\%$ compared with approaches of using FDR criteria only. We now have modified this statement and removed "extremely stringent".

Minor comments:

1. In the abstract, state which area of the brain was studied.

Response: we have added the brain area (Brodmann area 7) in the abstract.

2. Make it clear that SNCA is the gene encoding alpha synuclein-you do this later in the introduction but perhaps move the explanation to the first mention of the gene name.

Response: Thanks. We have moved the explanation to its first place.

3. "toxin like" should be "toxin-like"

Response: Good catch. We have revised and also checked for further grammatical errors throughout.

4. "Epigenetic mechanisms including the methylation of CpG dinucleotides on genomic DNA, mediate the gene-environment interactions thought to be involved in the development of the disease"

This sentence is ambiguous: it can either be read as a claim that epigenetic mechanisms mediate the gene-environment interactions that are believed to be involved in DLB (which is quite a strong claim that would definitely need a supporting reference) or that epigenetic mechanisms mediate gene-environment interactions in general, which is, of course, a widely accepted belief. Please make it clear which meaning is correct.

Response: We have revised it to (line 69-71):

“Epigenetic mechanisms including the methylation of CpG dinucleotides on genomic DNA are known to mediate the gene-environment interactions, and are thought to be involved in the development of the disease.”

5. “Compared with other neurodegenerative diseases such as PD and AD, where comprehensive studies of DNA methylation regulation have been conducted, very few studies have undertaken this for DLB”
It is true that comprehensive studies of DNA methylation have been carried out for these conditions; however, these studies have generally just identified sites showing differences in methylation rather than showing that the DNA methylation is actively involved in regulating anything. As such, the term “DNA methylation regulation” is misleading. In many cases, the role of DNA methylation is unknown and, in many places in the genome, variation in methylation does not appear to be correlated with gene expression.

Response: We are in total agreement with the reviewer that we cannot infer the “DNA methylation regulation. We now have changed the “regulation” to “dynamics” (line 77).

6. “Desplats et al. (2011) found global DNA hypo-methylation...”: clarify whether this is the SNCA gene or the whole genome.

Response: Desplats et al. observed a global DNA hypo-methylation using ELISA. So their result was for the whole genome. However, the authors also observed a hypo-methylation at intron 1 of the *SNCA* gene using methylation-sensitive PCR.

7. “ANK1, a well-known epigenetic target of AD”: not accurate to call ANK1 a target of AD as this suggests a direction of effect that is not known.

Response: We agree the causality of ANK1 on AD is not confirmed. We have revised to “ANK1, a gene previously reported to be associated with AD” (line 90).

8. “Gender” should be “sex”

Response: We have replaced “gender” with “sex” for the whole manuscript.

9. “Probes when the last 3 bases in its target sequence overlapped with known single nucleotide polymorphisms (SNPs)”: did you only consider SNPs with minor allele frequency above a certain threshold? Is there a study that supports the use of the three base cut-off? I am aware of the recommendation to use a five base cut-off by Zhou et al. (2017). This sentence also contradicts the first sentence in the “Differential methylation site analysis” section of the results section, where you state “Following the removal of all probes which overlapped with known single nucleotide polymorphisms”. Please clarify the approach used.

Response: In our previous version, we adopted the default parameter of an overlap of 3 bases in the RnBeads package. There are different options including 5 bases or “any” when some bases in the target sequence overlap with one or more SNPs. The choice of threshold might be dataset- or biological question-specific, leaving it challenging to determine the optimal threshold. We apologize for the misleading of the sentence “Following the removal of all probes which overlapped with known single nucleotide polymorphisms.” We actually meant that we removed probes that were overlapping with known SNPs in their target sequences (i.e., not only the exact position overlaps).

In our new version, to be more conservative on removing the genetic influences on the DNA methylation changes, we choose to remove all the probes when any bases in their target sequence overlaps with a SNP regardless of the minor allele frequency. We note that this process might lead to false negatives.

Nevertheless, as we don't have access to a replication cohort at this time to validate our discovery, we choose to report the DMCs or DMRs in a more conservative way.

10. "overlapped with non-CpG context": can you clarify what this means?

Response: The EPIC array also captures non-CpG or CpH sites including CpA, CpC, and CpT etc. We have detailed the number of CpHs in the main text.

11. "probes with unreliable measurements (p-value>0.05)": clarify that this is the detection p-value

Response: Yes, this is indicating the detection p-value. We have clarified it in the main text.

12. "along with those probes previously described to hybridize to multiple locations in the genome were removed". Please cite the reference you used to identify these probes.

Response: The cross-reactive probes were downloaded from the supplementary Table 1 of Pidsley et al.'s study. We have added the relevant reference as below (line 144):

Pidsley, R. et al. Critical evaluation of the Illumina MethylationEPIC BeadChip microarray for whole-genome DNA methylation profiling. *Genome Biology* 17, (2016).

13. Table 1: add the unit of measurement for post-mortem interval

Response: Thanks for your comment. We have added the unit for PMI (hour) in **Table 1**.

14. "These DMCs showed a mild-to-moderate effect size with the average absolute β -difference of 0.07 (sd= 0.021, range from -0.18 to 0.16)". I am not convinced that this is a useful statement. I don't think an average beta difference is a particularly useful figure to report, particularly when the causes and consequences of variation in DNA methylation are poorly understood and appear to vary according to context (i.e., a small difference might be very important in some contexts whilst a large difference might have little effect in others). It's also worth noting that that effect of a given change in beta-value will quite possibly vary depending on where on the scale of possible beta-values this change occurs (i.e., where between 0 and 1).

Response: We agree with the reviewer on this. We reported the average methylation level per group and average methylation differences as we just wanted to inform the scientific community the kind of "landscape" of methylation changes in DLB. This information may also prove useful for power estimation in future studies.

15. It would be preferable for the figures in Figure 1A to be slightly larger

Response: We have split **Figure 1** into two figures. We added a Q-Q plot as Figure 1A, forming a new **Figure 1**, and moved the remaining elements to **Figure 2**.

16. Table 2 should be ordered on p-value as there are ties between the FDR adjusted p-values. Gene names should be italicised (in other tables too).

Response: We have revised all the tables following your suggestion.

17. Some words are unnecessarily capitalised (bivalent, zinc)

Response: We now have revised these words.

18. Please clarify the meaning of this: "...the majority were not captured by marginal DLB associations and may represent important, additional loci for DLB related epigenetic changes"

Response: Sorry for the confusion. The WGCNA analysis mainly revealed highly correlated CpG modules which are correlated with the DLB/control phenotype regardless of their statistical significance in the

EWAS tests. Most of the modules we detected do not overlap with the inferred significant DLB-associated DMCs, thus we thought that this type of CpG-module might provide additional information related to DLB. To avoid this confusion, we have deleted this sentence in the new version.

19. Use “methylome-wide association study” or “epigenome-wide association study” rather than “genome-wide association study”

Response: Thanks for your suggestion. We have revised these phrases in the text.

20. The meaning of “intelligence related diseases” is not clear

Response: This points to the disease term “Intelligence” in the DisGeNet database with UMLS CUI of C0021704 and MeSH Class of Behavior and Behavior Mechanisms (MeSH:D007360), see below the snapshot from the DisGeNet database webpage:

Intelligence
Name: Intelligence
UMLS CUI: C0021704
Type: phenotype
MeSH Class: Behavior and Behavior Mechanisms
MeSH: D007360
OMIM: None
Semantic Type: Mental Process
Phenotypic abnormality: None
Disease Ontology: None

In the new version of this study, we have updated the WGCNA analysis and similar disease terms were enriched, such as mild cognitive disorder, memory impairment etc., for certain modules.

21. “Our study corroborates some previously reported genes that were associated with DLB”. Explain how these genes have been implicated in DBP (i.e. GWAS or EWAS etc)

Response: Sorry for the confusion. These previously-reported genes were identified through different epigenetic analyses, as indicated in a recent review paper (Urbizu et al. *Int. J. Mol. Sci.* 2020) – most of them were found through low-throughput pyrosequencing (Ozaki, Y. et al. *Acta Neurol. Scand.* **2020**; Tsuchida, T. et al. *Brain Res.* **2018**; Liu, H.C. et al. *Neurosci. Lett.* **2008**; Tulloch, J. et al. *Alzheimer’s Dement.* **2018**) and few of them were based on genome-wide DNA methylome such as WGBS or GoldenGate Assay (Sanchez-Mut, J.V. et al. *Transl. Psychiatr.* **2016**; Fernandez, A.F. et al. *Genome Res.* **2012**),. There were no large-scale EWAS studies conducted so far except Pihlstrøm et al.’s study which is a preprint.

We have deleted this sentence in the discussion section but instead adding the following sentences to the last paragraph of the “Differential methylation site analysis” section in result (line 287-293):

“Most of them were found through low-throughput pyrosequencing or Methylation-specific PCR (e.g. APOE, CRY1, DRD2, FGFR3, PER1, SNCA, and SELENOW)^{25–28,47,48} and only few of them were based on genome-wide DNA methylome such as WGBS or Illumina Array (e.g. ANK1, NBL1, and PTK6)^{29,49}”.

22. The findings re. PAK6 are interesting: is there any evidence linking altered methylation at the associated sites to an effect on expression of PAK6?

Response: We are not aware of any gene expression data in the same region of the brain for DLB and

control samples, and thus we have no direct evidence to link the hypomethylation of PAK6 in DLB patients to the gene expression changes of PAK6 in the same patients. This is worth exploring in the future.

23. “To the best of our knowledge, this is the first reported DNA methylation association analysis to have been conducted on DLB brain tissue”. How does this relate to the study of Pihlstrøm et al., which is also discussed?

Response: During the submission of our study, we noticed that the study of Pihlstrøm et al., was uploaded to medRxiv. Together with the Pihlstrøm et al., study, we are at the first wave of DLB EWAS studies. We now have modified the following statement (line 420-423):

“To the best of our knowledge, this is one of the first couple of methylome-wide association analysis studies to have been conducted on DLB brain tissue.”

24. The bias towards finding hypomethylation in the DLB cases is an interesting finding. How does this relate to the findings of Pihlstrøm et al.? Are there any studies of gene expression in DLB (preferably brain) that also show a bias in the direction of the associated gene expression changes?

Response: In a blood DNA methylation paper (Nasamran et al. *Alzheimer’s Dement.*, 2021), reduced DLB methylation compared with Parkinson’s disease dementia (PDD) was reported. Also, hypomethylation was observed in multiple genes including *SNCA*, *APP* etc. in other DLB studies. These observations suggest that reduced methylation at certain genes might be associated with DLB pathogenesis and/or progression. It would be ideal to have gene expression data to support or validate the EWAS discoveries. However, unfortunately, no such gene expression data (particularly in the same brain tissue) are available currently for integrative analysis. This will be a very interesting direction for future investigations.

Nasamran, CA, Sachan, ANS, Mott, J, et al. Differential blood DNA methylation across Lewy body dementias. *Alzheimer's Dement.* 2021; 13:e12156. <https://doi.org/10.1002/dad2.12156>

REVIEWERS' COMMENTS:

Reviewer #1 (Remarks to the Author):

The authors have put in considerable work to address concerns with this manuscript. Given the issue related to running DLB and control samples being run on different batches of arrays it is unlikely that removing batch effects can be achieved without removing disease related signal. This would also explain how lambda values are still inflated (1.1) even with use of PCs. In my opinion the authors have done all they can given this design error and low sample number and have also highlighted these limitations in the manuscript. I would recommend this paper for publication despite findings having considerable caveats, subject to the following minor change:

1. Line 88 – authors suggest that previous work (reference 29) had identified 1075 CpGs associated with DLB, in truth there is 1075 DMRs that are associated with expression of genes not necessarily DLB specific. Coupled with only being generated from 1 case vs 1 control RRBS data means this statement is rather misleading.

Reviewer #2 (Remarks to the Author):

The authors have addressed the concerns and comments.

Reviewer #3 (Remarks to the Author):

The authors have carried out a thorough revision of the paper and they have satisfactorily addressed my comments. I have just one minor comment:

“Specifically, profiling was undertaken in post-mortem brain tissue (neocortex, Brodmann area 7)”

The neocortex encompasses many areas, please be more specific: Brodmann area 7 is parietal cortex.